# Validation of DBFOLD: An efficient algorithm for computing folding pathways of complex proteins

**Amir Bitran** [1,2]*, **William M. Jacobs** [3], **Eugene Shakhnovich** [1]

**1** Department of Chemistry and Chemical Biology, Harvard University, Cambridge, Massachusetts, United States of America, **2** Harvard University Program in Biophysics, Harvard University, Cambridge, Massachusetts, United States of America, **3** Department of Chemistry, Princeton University, Princeton, New Jersey, United States of America

* bitranamir@gmail.com

**Data Availability Statement:** All extracted simulation data related to protein G is available within the GitHub repository linked to in the paper, specifically within the 1igd_data subdirectory:

## Abstract

Atomistic simulations can provide valuable, experimentally-verifiable insights into protein folding mechanisms, but existing *ab initio* simulation methods are restricted to only the smallest proteins due to severe computational speed limits. The folding of larger proteins has been studied using native-centric potential functions, but such models omit the potentially crucial role of non-native interactions. Here, we present an algorithm, entitled DBFOLD, which can predict folding pathways for a wide range of proteins while accounting for the effects of non-native contacts. In addition, DBFOLD can predict the relative rates of different transitions within a protein's folding pathway. To accomplish this, rather than directly simulating folding, our method combines equilibrium Monte-Carlo simulations, which deploy enhanced sampling, with *unfolding* simulations at high temperatures. We show that under certain conditions, trajectories from these two types of simulations can be jointly analyzed to compute unknown folding rates from detailed balance. This requires inferring free energies from the equilibrium simulations, and extrapolating transition rates from the unfolding simulations to lower, physiologically-reasonable temperatures at which the native state is marginally stable. As a proof of principle, we show that our method can accurately predict folding pathways and Monte-Carlo rates for the well-characterized Streptococcal protein G. We then show that our method significantly reduces the amount of computation time required to compute the folding pathways of large, misfolding-prone proteins that lie beyond the reach of existing direct simulation. Our algorithm, which is available online, can generate detailed atomistic models of protein folding mechanisms while shedding light on the role of non-native intermediates which may crucially affect organismal fitness and are frequently implicated in disease.

## Author summary

Many proteins must adopt a specific structure in order to function. Computational simulations have been used to shed light on the mechanisms of protein folding, but

https://github.com/amirbitran/dbfold/tree/master/1igd_data Instructions for how to analyze this data are provided in dbfold_test.ipynb within the master directory. All data related to other proteins presented in main text figures can be found at: https://figshare.com/articles/Analyzed_data/11496954.

**Funding:** AB was funded by the National Science Foundation GRFP (DGE1745303) and the Harvard Molecular Biophysics Training Grant (PI: James M Hogle, NIH/ NIGMS T32 GM008313). WMJ was funded by NIH grant F32GM116231. ES was funded by NIH grant R01 GM124044. The funders had no role in study design, data collection and analysis, decision to publish, or preparation of the manuscript.

**Competing interests:** The authors have declared that no competing interests exist.

unfortunately, realistic simulations can typically only be run for small proteins, due to severe limits in computational speed. Here, we present a method to solve this problem, whereby instead of directly simulating folding from an unfolded state, we run simulations that allow for computation of equilibrium folding free energies, alongside high temperature simulations to compute unfolding rates. From these quantities, folding rates can be computed using detailed balance. Importantly, our method can account for the effects of nonnative contacts which transiently form during folding and must be broken prior to adoption of the native state. Such contacts, which are often excluded from simple models of folding, may crucially affect real protein folding pathways and are often observed in folding intermediates implicated in disease.

## Introduction

Many proteins suffer from very slow or inefficient folding from a denatured state owing to a tendency to misfold into non-native intermediates. Such intermediates can be detrimental *in vivo*, where they may be degraded, form toxic oligomers, or aggregate, potentially leading to loss of fitness and/or disease [1–7]. Organisms deploy various cellular mechanisms to mitigate protein misfolding including chaperones [4, 8–10] and co-translational folding on the ribosome [5, 11–17], which may be enhanced by slowly translating codons located at nascent chain lengths that show optimal folding properties [15–17]. Despite the fact that non-native folding intermediates exert widespread and significant consequences, we have yet to develop a detailed atomstic understanding of how they slow folding, and how cellular mechanisms reduce their formation and detrimental effects. All-atom simulation methods such as Molecular Dynamics (MD) and Monte-Carlo (MC) simulations have the potential to generate detailed models of folding, but unfortunately their use has thus far been restricted to small proteins which typically engage in few nonnatives interactions ($< \approx 100$ amino acids), due to severe limits in computational speed [18]. The folding of larger proteins, which comprise the majority of the proteome, can be simulated using native-centric Go models [18–20], but such models lack non-native interactions which may crucially affect real folding pathways.

To address these difficulties, various enhanced sampling techniques have been developed that allow the folding of complex proteins to be investigated without requiring *ab initio* simulation. For instance, replica exchange or parallel-tempering [21] whereby multiple simulations are run in parallel under different conditions and information is periodically exchanged between cores, can assist a protein in sampling folding intermediates that are separated from the initial structure (often the equilibrated native state) by large kinetic barriers. Such simulations can then be analyzed using methods such as WHAM [22] or MBAR [23] to infer the free energies of intermediates. However, the implementation of replica exchange comes at the expense of realistic state transition kinetics, and replica exchange is unlikely to promote sampling saddle points in the free energy landscape, thus hindering barrier-height computation, Biasing techniques such as umbrella sampling [24, 25] and Metadynamics [26] can improve sampling along saddle points, but they are only useful if proper order parameters or collective variables along which slow transitions occur are known in advance, which is not the case for most proteins. Other sampling techniques such as transition-path sampling [27, 28] and forward flux sampling [29] are more tolerant of uncertainty in the order parameter(s), but these are extremely computationally expensive to implement for large proteins with multiple intermediates.

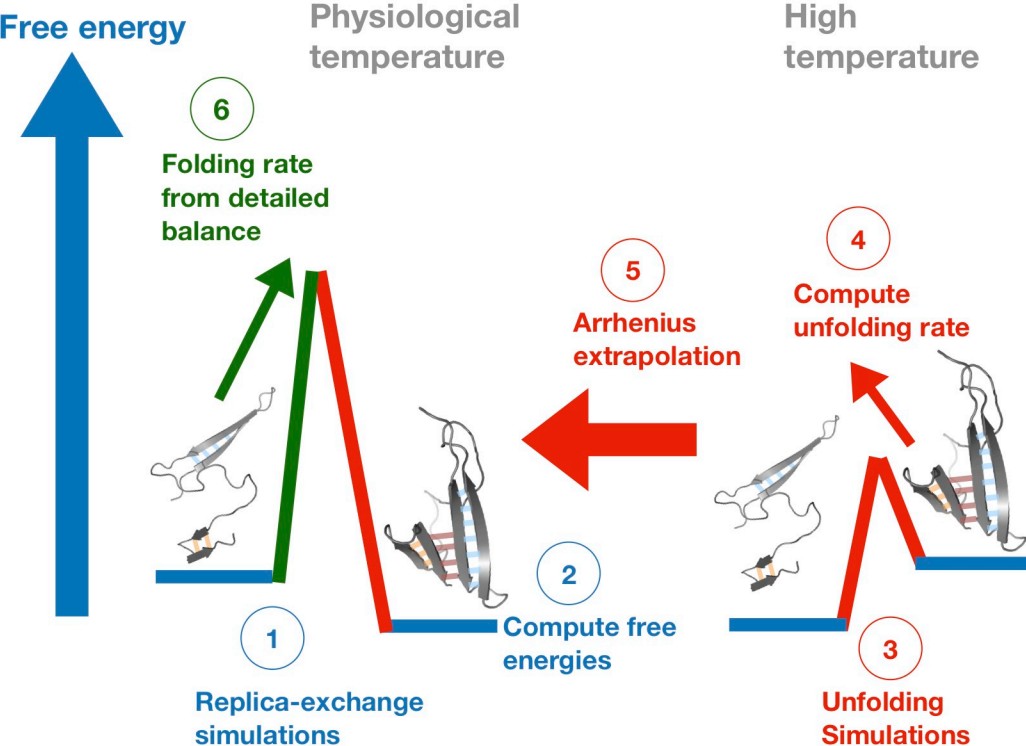

**Fig 1. Schematic overview of method for computing folding rates.** 1. All-atom replica-exchange simulations with umbrella sampling are run at a wide range of temperatures spanning physiologically-reasonable ones. 2. A folding landscape is defined by identifying coarse-grained intermediates, which may be significantly stabilized by non-native contacts (see text), and the relative free energies of these intermediates are computed based on simulations in previous step. 3. High-temperature unfolding simulations are run without replica exchange nor umbrella biasing. 4. Sequential rates of unfolding between progressively-less folded intermediates are computed from unfolding simulations. 5. These unfolding rates are extrapolated down to physiological temperatures using the Arrhenius equation. 6. Unknown folding rates are computed from unfolding rates (obtained in step 5) and state free energies (obtained in step 2) using the principle of detailed balance. For details regarding these steps, see the main text and Materials and methods.

Here, we develop a computational method that allows for the prediction of realistic folding intermediates and transition rates for a wide range of proteins without resorting to direct *ab initio* simulation nor computationally expensive sampling methods (Fig 1). In essence, our method combines enhanced sampling techniques, specifically replica exchange and umbrella biasing, with high temperature unfolding simulations, Under certain conditions described in the next section, unfolding rates from these latter simulations can be extrapolated to physio-logically-reasonable temperatures at which the native state is stable, and combined with inter-mediate free energies inferred from the former simulations to compute unknown folding rates from detailed balance. In what follows, we develop theory to elucidate the conditions under which our method can be applied. As a proof of principle, we then apply this technique to investigate the folding pathways, including the role of nonnative states, for the well-character-ized Streptococcal protein G. We show that we can accurately predict transition rates between protein G folding intermediates, which for this small protein, can be verified via direct folding simulation. Finally, we discuss how this method can be applied to larger, more complex pro-teins whose folding pathways have not been well studied. Our implementation of this algo-rithm, DBFOLD, includes both the latest version of MCPU–an all-atom Monte-Carlo simulation platform that we use in this work–as well as a user-friendly Python package that analyzes simulations to compute folding rates using the techniques described here. In its

current implementation, DBFOLD computes rates in Monte-Carlo (MC) units. This allows for meaningful comparison of the relative rates of different steps in a given protein's folding pathway. Additionally, MC folding rates can be meaningfully compared across truncated forms of a given protein in order to elucidate how vectorial synthesis affects co-translational folding [17].

## Results

### Developing a coarse-grained folding landscape

In order to apply this technique, we must first coarse-grain a protein's folding landscape into a set of meaningful intermediates. It is crucial that this be done carefully such that detailed balance can be used to compute folding rates between the resulting intermediates–this may not be the case, for instance, if intermediates are defined in such a way that transitions are non-Markovian. To proceed, we deploy an approach similar to the one described in [20], where we generate a native contact map for an equilibrated protein and identify islands of contiguous native contacts, referred to as *substructures* (See Fig 2A for examples, and Materials and methods for details). We expect that during each on-pathway transition in the folding/unfolding process, one such substructure forms/breaks cooperatively, and that these transitions are accompanied by a high free energy barrier [20]. This is justified because making the first set of contacts in a substructure typically entails the formation of a loop, which carries a large entropic loss that is not compensated by an enthalpic benefit until subsequent contacts within the substructure form. We next define a *topological configuration* as a possible subset of native substructures that can be formed during the folding process. In Fig 2B, we show sample structures of *E. Coli* DHFR assigned to various topological configurations, for example *abcdefg* (all native substructures formed), *cd* (only substructures *c* and *d* formed), and $\emptyset$ (no substructures formed).

In the simple case where only native contacts can form during folding, then we expect that transitions between topological configurations will show Markovian dwell-time distributions, owing to the high free energy barriers associated with the transitions. The resulting network of topological configurations thus resembles a Markov state model [30] in which states are defined according to structural similarity, rather than based on kinetic data. But in reality, a protein may also form nonnative contacts at any stage in the folding process which may impede the formation of additional native substructure(s), and must be broken before productive folding can proceed. We define a *coarse state* $S_i = \{s_i^n\}$ as the collection of all microstates containing nonnative contacts that are topologically consistent with a given topological configuration, indexed by $i$, as well as microstates with topological configuration $i$ that lack nonnative contacts. The presence of nonnative microstates may lead to non-Markovian behavior for transitions between $S_i$ and some other coarse state $S_j$ with topological configuration indexed by $j$ (which differs from the one indexed by $i$ by the formation/breaking of one substructure), owing to complex internal dynamics involving these microstates. Nonetheless, it turns out that, so long as certain conditions are satisfied, then detailed balance can still be used to compute unknown transition rates between coarse states. Two such non-mutually exclusive conditions are briefly described below, and detailed in the S1 Text section "Details on conditions for applicability of method".

### Conditions when detailed balance can be used to compute folding rates

**Condition I**: Let us assume that all microstates $s_i^n \in S_i$, including those with nonnative contacts, equilibrate rapidly with each other relative to the fastest timescale of transition to any

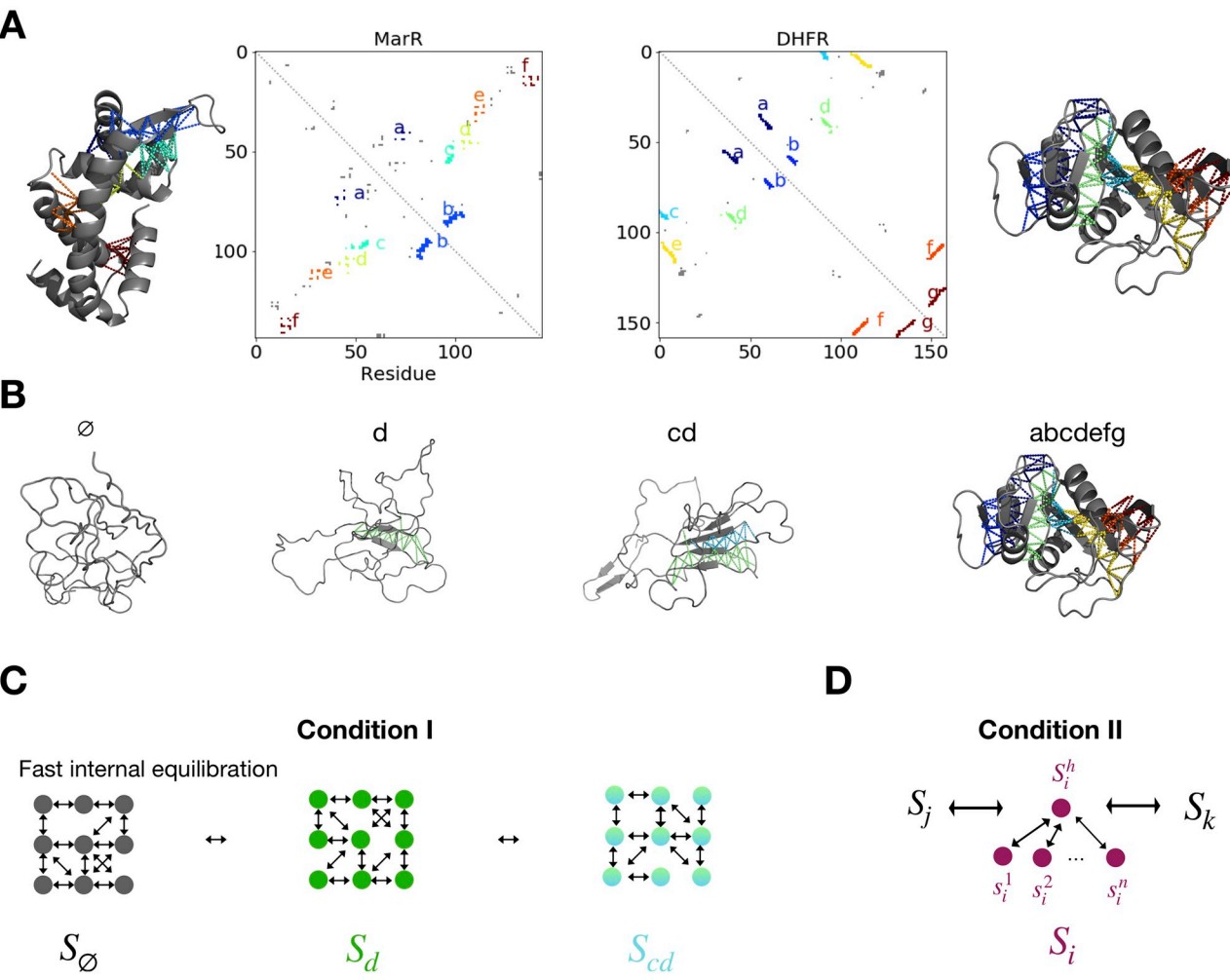

**Fig 2. Coarse-graining the folding landscape.** Examples of native contact maps alongside native structures with highlighted substructures (islands of continuous native contacts, see main text and Materials and methods) for the *E. Coli* proteins MarR (left) and DHFR (right). Substructures are labeled alphabetically. Native contacts not assigned to a substructure are shown in gray on the contact maps. We note that, for brevity and ease of visualization, we omit contacts involving residues less than 8 amino acids apart in sequence (and thus intra-helical contacts are excluded). (B) Sample topological configurations, alongside representative simulation snapshots for DHFR are shown. A snapshot is assigned to a given topological configuration if it contains a certain subset of folded native substructures (indicated by the labels above, see Materials and methods for details regarding the assignment process), and configuration ∅ includes snapshots with no folded native substructures. (C-D) Schematic illustrations of conditions I (C) and II (D) under which a detailed-balance like relationship can be used to compute folding rates between topological configurations given equilibrium probabilities (from enhanced sampling) and unfolding rates (from high-temperature simulations). Circles represent microstates consistent with a given coarse-state, which may contain nonnative contacts, while double-arrows represent transitions. For details on these conditions, see the main text.

other coarse state $S_j$. This assumption is often valid as nonnative contacts are frequently shorter in range than native substructures and are thus expected to form rapidly relative to native substructures. This condition results in a separation of timescales such that $S_i$ shows approximately Markovian behavior at timescales significantly greater than the internal equilibrium time (Fig 2C). Letting $H(i, j)$ denote the Hamming distance between two topological configurations (i.e. the number of substructures by which they differ), then the probability $P_t(S_i)$ of occupying $S_i$ as a function of time will approximately satisfy the master equation:

$$\frac{d}{dt}P_t(S_i) = \sum_j \delta_{H(i,j),1} \; k_{j \to i} P_t(S_j) - \left( \sum_j \delta_{H(i,j),1} \; k_{i \to j} \right) P_t(S_i) \tag{1}$$

Where $\delta_{H(i,\,j),1}$, the Kronecker delta function, has value 1 if $i$ and $j$ differ by exaclty one sub-structure and 0 otherwise, while $k_{i\to j}$ and $k_{j\to i}$ refer to the rates of transition from $S_i$ to $S_j$ and vice versa, respectively. These rates are generally temperature-dependent as discussed later, but the temperature-dependence is omitted from the notation for brevity. Importantly, these transition rates satisfy detailed balance. That is, letting $P_{eq}(S_i)$ and $P_{eq}(S_j)$ denote the equilibrium Boltzmann probabilities and $G_{S_i}$ and $G_{S_j}$ the free energies of states $S_i$ and $S_j$, respectively, we have:

$$\frac{k_{i\to j}}{k_{j\to i}} = \frac{P_{eq}(S_j)}{P_{eq}(S_i)} = e^{(G_{S_i}-G_{S_j})/k_B T} \tag{2}$$

Fortunately, free energy differences between pairs of coarse states $S_i$ and $S_j$ can be accurately inferred from simulations with enhanced sampling so long as these coarse states are sufficiently sampled at equilibrium such that statistical uncertainties are low. Furthermore, if we assume without loss of generality that topological configuration $j$ is more folded (i.e. contains one more formed substructure) than $i$, then we can, under certain reasonable conditions, extrapolate the unfolding rate $k_{j\to i}$ from simulations run at high temperature (See subsection Requirements for Extrapolation below). Thus, using Eq (2), we can solve for the unknown folding rate $k_{i\to j}$.

We further note that the folding landscape can be further coarse-grained by grouping together sets of coarse states into a *cluster* $C_k$, so long as the slowest timescale of exchange between states within $C_k$ is significantly faster than the fastest timescale of exchange between $C_k$ and any other cluster $C_l$. If so, then these clusters will themselves behave as Markovian states whose occupancy probabilities satisfy Eqs (1) and (2). Such clustering may be desirable so as to reduce the number of parameters in the model, as explored for protein G below.

**Condition II**. In case condition I is not satisfied for a coarse state $S_i$, then the dwell-time distribution associated with the transition to any other coarse state $S_j$ will show multi-exponential behavior, and thus we cannot meaningfully define a single rate $k_{i\to j}$ for the transition. Nonetheless, it turns out a detailed-balance like relationship can still be used to compute the *mean first passage time* (MFPT) to state $S_j$ so long as certain conditions are satisfied. Namely, we consider a subset $S_i^h \subset S_i$, termed a *hub state*, which contains the only microstates belonging to $S_i$ from which transitions to $S_j$ can occur (Fig 2D). We require that 1.) All $s_i^n \in S_i^h$ equilibrate with one another rapidly relative to transitions to $S_j$, such that the hub itself satisfies condition I and shows Markovian behavior, and 2.) Upon first reaching $S_i$, the system must start in the hub with probability 1. Under these conditions, we can compute the MFPT to reach $S_j$, conditioned on the facts that the protein has just transitioned into $S_i$ (and thus, by construction, currently resides within $S_i^h$) and does not first reach any other coarse state $S_k$. By considering a modified version of the master equation and computing the inverse of this mean first passage time $<\tau>_{i\to j}$ (S1 Text), we obtain:

$$\frac{1}{<\tau>_{i\to j}} = \frac{P_{eq}(S_j)}{P_{eq}(S_i)}\frac{1}{<\tau>_{j\to i}} = \frac{e^{(G_{S_i}-G_{S_j})/k_B T}}{<\tau>_{j\to i}} \tag{3}$$

Where we have assumed that the reverse transition from $S_j$ into $S_i$ also satisfies either condition I or II. We thus find that, even though the $S_i$ to $S_j$ transition does not show Markovian behavior, the inverse MFPTs nonetheless satisfies a relationship akin to Eq (2). Thus, so long as we can extrapolate the reverse unfolding timescale $<\tau>_{j\to i}$ from high temperatures (justifiable under conditions described below), then we can use Eq (3) to compute the folding the MFPT as a characteristic timescale for the folding transition between coarse states $S_i$ and $S_j$. As in the

previous section, we can cluster $S_i$ with any other coarse states with which it rapidly exchanges, so long as these coarse states satisfy either conditions I or II. If so, then the resulting cluster's hub state retains the properties above and thus the cluster still satisfies condition II. Finally we note that whereas condition I applies to a coarse state/cluster as a whole, condition II is specific to a *transition*, and may not apply to an entire state/cluster. For example, suppose the subset of microstates via which $S_i$ can transition to $S_j$ does not fully overlap with the subset that permits transition to $S_k$. If we further suppose that only the subset that allows transition to $S_j$ rapidly equilibrates internally, then the $S_i$ to $S_j$ transition will satisfy condition II, but the $S_i$ to $S_k$ transition will not.

**Requirements for extrapolation.** Under certain conditions, unfolding rates obtained from high temperature simulations can be extrapolated using a an appropriate model. In this work, we use the Arrhenius equation, given by

$$k_{j \to i}(T) = k_{j \to i}^0 e^{-\Delta E_{j \to i}^{\ddagger}/k_B T} \tag{4}$$

Where $k_{j \to i}(T)$ is the transition rates from clusters $C_j$ to $C_i$ as in the previous section (with the temperature dependence now explicitly considered), $k_{j \to i}^0$ is an intrinsic, temperature-independent rate constant and $\Delta E_{j \to i}$ is the activation energy for this transition. The Arrhenius equation is used here because, in Monte-Carlo simulations, intrinsic rates of molecular motion (and by extension, $k_{j \to i}^0$) do not depend on temperature. However, in molecular dynamics simulations, alternative models such as the the Eyring equation may be more appropriate in certain contexts. But in either case, certain requirements must be satisfied for these equations to be valid. For concreteness, let us assume that a transition from clusters $C_j$ to $C_i$ involves the disruption of some native substructure $s$. Then we require that:

1. The breaking of $s$ must involve crossing a single large barrier, which is generally expected for substructure disruption as discussed earlier.

2. The position of the saddle in the free energy landscape for the breaking of $s$ must not change over our temperature range of interest. Typically, this saddle will occur when only one or a few contacts belonging to $s$ are formed. At this point, most of the enthalpy that stabilizes $s$ will have been lost, but the entropy associated with disrupting $s$ will not yet have been gained, as the residual contacts will severely restrict the conformational freedom of residues involved in this substructure. However, the precise position at which this saddle occurs may change over a large temperature range.

Moreover, in some cases unfolding of $s$ will be preceded by the breaking of some set of non-native contacts $n$ which are observed with high probability in cluster $C_j$, but not in cluster $C_i$. This may occur, for instance, if $n$ and substructure $s$ are energetically coupled. If so, then we additionally require that

3. Cluster $C_j$ must satisfy conditions I or II. In case condition II, but not I is satisfied, we cannot define a single unfolding rate, and we instead extrapolate the inverse mean-first passage time (MFPT) to unfolding. If condition I is satisfied, then this inverse MFPT is equivalent to the unfolding rate. Thus for generality we deal with with the inverse MFPT throughout rest of this text

4. In case $C_j$ is composed of multiple nonnative microstates $s_j^n$ (all of which contain nonnative contacts that must be broken), then one such microstate $s_j^m *$ must show a significantly greater probability of equilibrium occupancy than the others at all temperatures of interest. The presence of multiple minima with comparable but non-identical equilibrium

probabilities will produce nonlinearities in the dependence of $\log(k_{j \to i})$ on inverse temperature (See SI section with heading "Justification of Arrhenius kinetics") as different minima may be favored at different temperatures.

We further note that, when utilizing this method in practice, we assume that the protein will fold via the opposite sequence of substructures as that through which it unfolds at high temperatures, as we can only obtain mean-first passage times (MFPTs) to folding for transitions for which we have extrapolated the reverse unfolding MFPT. This is generally true because a protein will transition in both directions via whichever sequence of topological configurations involves the lowest rate-limiting barrier, but this optimal sequence may change over a wide temperature range.

## Computing equilibrium folding properties for protein G

As a proof of principle, we now apply our method on Streptocaccal Protein G, a model protein whose folding has been extensively studied using both computational and experimental methods [20, 31–38]. We begin by running equilibrium simulations with replica exchange and umbrella sampling using native contacts as the reaction coordinate along which we bias [39] (See Materials and methods). These simulations were run for a total of 1.2 billion Monte-Carlo (MC) steps, at which point convergence was reached (see Materials and methods and S1 Fig). This calculation required $\sim 1$ week of computation time on a cluster of 125 AMD Opteron 6376 CPUs. To compute equilibrium folding properties, we construct a coarse-grained folding landscape as described in the previous section and in Materials and methods (see Fig 3A). From the equilibrated protein G contact map, we identify the following substructures (which correspond precisely to those identified in [20]): Substructure *a* corresponds to the N-terminal beta hairpin, substructure *b* is the central helix, substructure *c* is the C-terminal beta hairpin, and substructure *d* is the parallel beta interface between the N and C-terminal hairpins. We then use the MBAR method ([23], see Materials and methods) to infer the potential of mean force (PMF) as a function of the fraction of native contacts, which allows us to compute a thermally-averaged melting curve (Fig 3B). This curve shows a cooperative transition, corresponding to the full denaturation of the protein, at the melting temperature $T = T_M$. To understand this transition in greater structural detail, we infer PMFs for each topological coarse state (Fig 3C) As expected, below $T_M$, the fully folded configuration (*abcd*) is lower in free energy than all others (Fig 3C, left), but as temperature is raised near the melting temperature, the folded state's free energy becomes comparable to that of less-folded configurations *b* and *bc* (Fig 3C, middle) while, well above the melting temperature, the fully unfolded state $\emptyset$ is favored (Fig 3C, right). We further note that at the lowest simulated temperature ($T \approx 0.45\ T_M$), only about 75% of native contacts are formed, owing to entropy. At temperatures below this, the fraction is expected to approach 1 as internal degrees of freedom become frozen. The beginnings of this gradual freezing transition are already apparent around $T \approx 0.45\ T_M$.

## Computing unfolding rates for protein G

To compute rates of transition between states, we run unfolding simulations at a range of high temperatures above $T_M$. By extrapolating unfolding times to physiological temperatures, we can then compute folding times using detailed balance provided our coarse states satisfy conditions I or II (detailed in the previous section), which we verify later. As discussed previously, we expect that the unfolding pathways will correspond to the reverse of the folding pathways. We find that the protein unfolds via one of two parallel pathways (Fig 4A) in which either substructure *d* or *a* can unfold first, followed by the other, and finally *c* unfolds last. We now

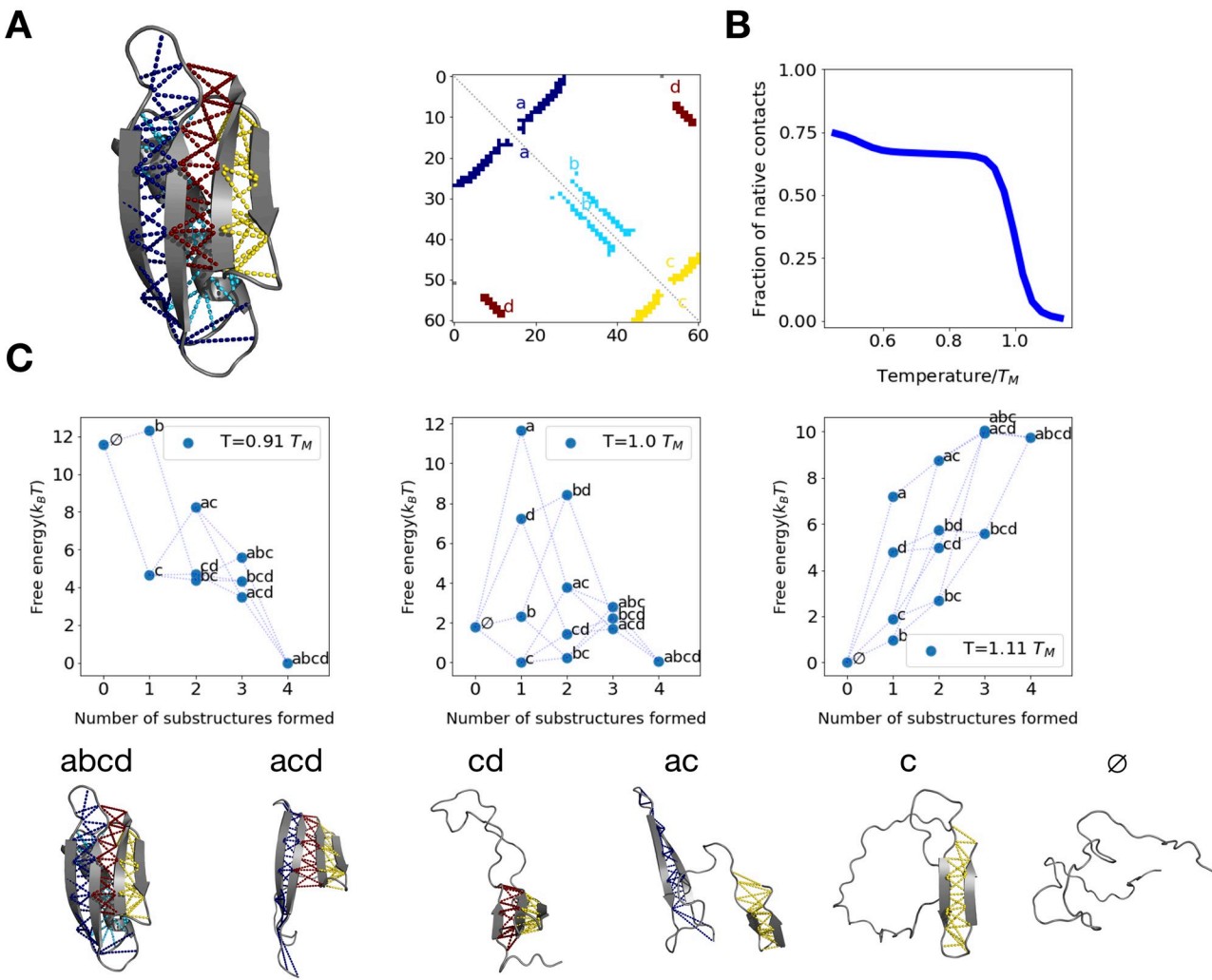

**Fig 3. Equilibrium folding properties for protein G.** (A) Equilibrated structure and contact map for protein G with each substructure highlighted. (B) Thermally-averaged equilibrium fraction of native contacts vs temperature for protein G. (C) Protein G potentials of mean force (PMFs) for each sampled *coarse state*, defined as a collection of microscopic configurations consistent with a given topological configuration (i.e. in which some subset of native substructures is formed). Each dot represents one such coarse state, whose topological configuration is indicated by its adjacent label. Coarse states are plotted at an x-value corresponding the number of formed substructures in their respective topological configuration, and are connected via dashed lines if their configurations differ by one substructure. Example simulation snapshots assigned to various topological configurations are shown below the plots.

attempt to simplify our model of folding by clustering together topological configurations that exchange rapidly (See Materials and methods). We find that the central helix (substructure *b*) folds and unfolds very rapidly compared to the timescale with which the beta-sheet substructures (*a*, *c*, and *d*) form/break. Thus, we construct kinetic clusters, which we refer to as *a(b)cd*, *a(b)c*, *(b)cd*, *(b)c*, and *b/∅*, in which every observed combination of beta substructures occurs alongside helix *b* in either its folded or unfolded state. So long as these clusters satisfy the requirements for extrapolation in the previous section, we expect to be able to extrapolate unfolding rates for each transition between clusters to low temperatures, and compute the reverse folding rates. Indeed, we find that the inverse mean-first passage times (MFPTs) for each unfolding step as a function of inverse temperature are well fit by the Arrhenius equation

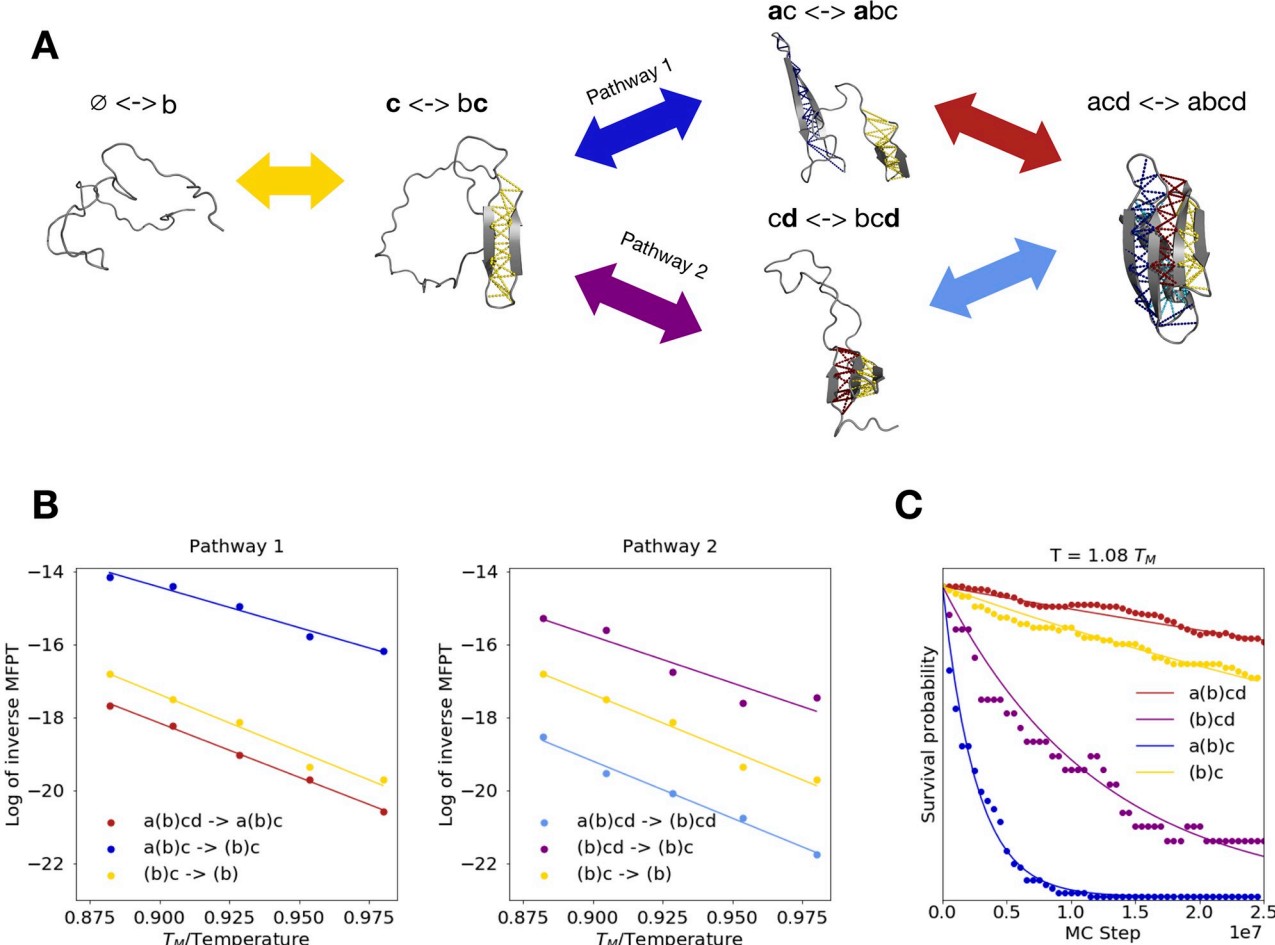

**Fig 4. Predicted unfolding pathways and extrapolating unfolding rates for protein G.** (A) Schematic for two predicted folding pathways, assumed to be the reverse of the observed dominant unfolding pathways. Topological configurations assigned to each cluster (with double arrows indicating that states are in fast exchange), are indicated along with sample snapshots. (B) Arrhenius plots show log of inverse MFPT to unfolding as a function of inverse temperature (normalized by the melting temperature $T_M$) for transitions associated with unfolding pathway 1 (left) and pathway 2 (right). Substructure $b$ is shown in parentheses in the cluster labels, indicating that each cluster contains two topological configurations: one in which $b$ is formed, and one in which $b$ is broken (as in panel (A)).(C) Survival probability as a function of Monte-Carlo (MC) time for each cluster (dots) alongside exponential fits (solid lines) during unfolding simulations at a temperature $T = 1.08\ T_m$ where $T_m$ denotes the melting temperature. Analogous plots at the other temperatures are shown in S3 Fig.

(Fig 4B). This suggests that unfolding times can be appropriately extrapolated down to physiologically-relevant temperatures. We note that all clusters show clear single-exponential survival probability curves as a function of MC step, with the exception of $(b)cd$ whose survival appears to show multi-exponential decay (Fig 4C and S3 Fig). This is because, as we show later, this latter cluster does not satisfy condition I. However, the $(b)cd \rightarrow (b)c$ unfolding transition nonetheless satisfies condition II, under which extrapolation of the unfolding MFPT is still possible as explained in the previous section. This is further supported by the fact this transition shows a reasonable Arrhenius fit (Fig 4B). Finally we note that a small amount of flux (<10% of trajectories) is observed to unfold through an alternative pathway whereby the N-terminus unfolds last, but these trajectories are excluded from Arrhenius fitting due to insufficient statistics.

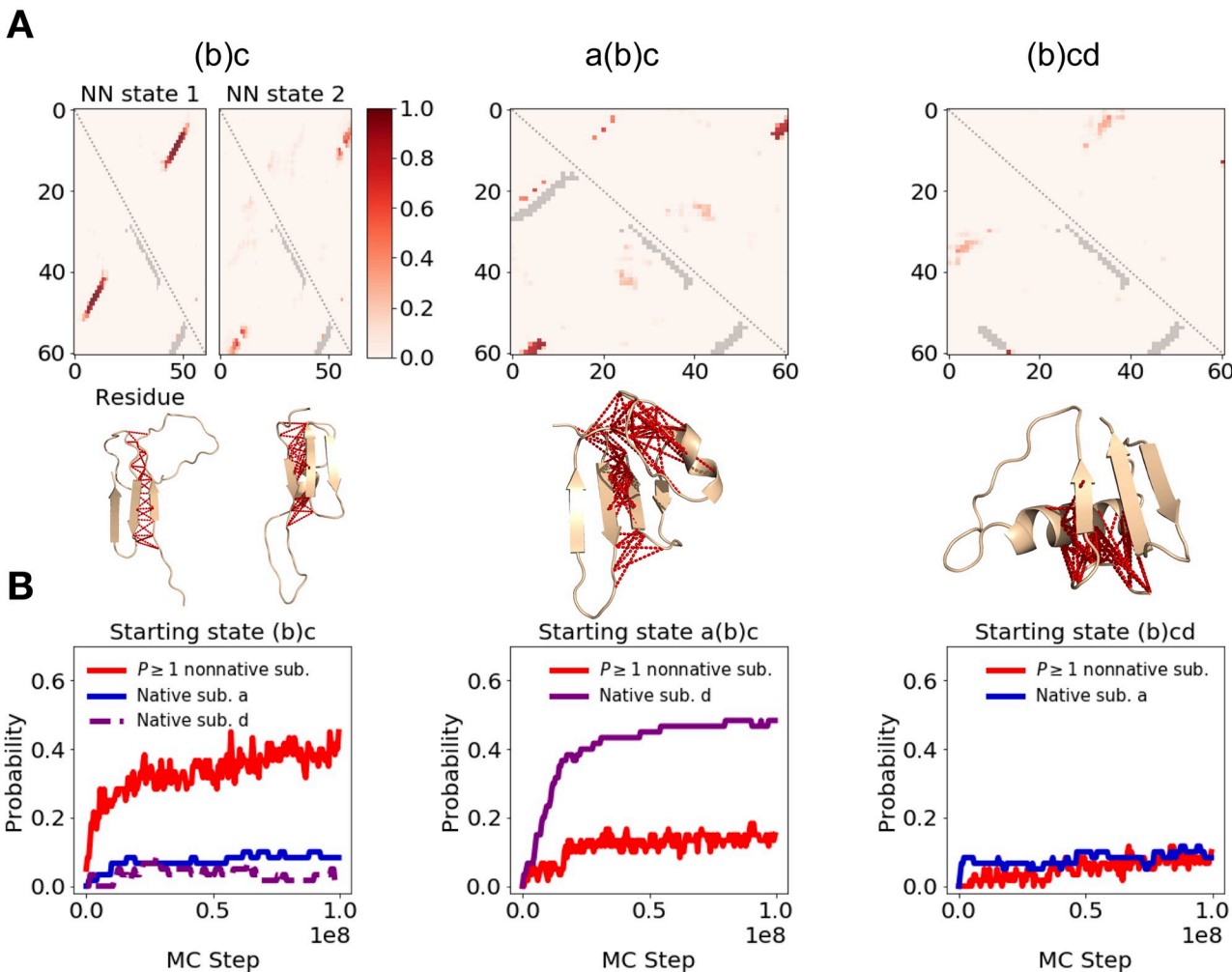

**Fig 5. Nonnative contacts that interfere with native protein G folding.** (A) Contact maps from equilibrium simulations are shown. Shades of red are used to indicate frequency with which nonnative contacts occur in simulation snapshots assigned to the cluster indicated above the respective map, drawn from a range of physiologically-reasonable temperatures around $T \approx 0.85\ T_M$, while gray contacts indicate formed native substructures. For cluster $(b)c$, the contact maps subdivide into two nonnative states, whereas one predominant state is observed for other configurations (See Materials and methods). Sample simulation snapshots assigned to each cluster are shown below the respective contact maps, with nonnative contacts indicated in red. (B) Refolding simulations at $T \approx 0.85\ T_M$ were run, initialized from snapshots assigned to the topological configurations shown above each panel but drawn from high temperatures such that almost no nonnative contacts are initially present. We plot, as a function of MC step, the probability of forming at least one nonnative substructure observed in replica simulations (as in panel A) and of forming native substructure $a$ (blue) or $d$ (purple) within these refolding simulations.

## Characterizing off-pathway nonnative states

Next, we characterize the off-pathway nonnative states the protein can adopt at each stage in the folding process. To this end, we generate nonnative contact maps for snapshots assigned to each cluster at physiologically-reasonable temperatures, then group these maps based on similarity to identify recurrent nonnative states (Fig 5A and S4 Fig, see also Materials and methods). We find that when the protein is fully unfolded (cluster $\emptyset/b$), it can form nonnative contacts, but these do impede the formation of the C-terminal beta hairipin, which is the first productive folding step (S4 Fig). In contrast, once that hairpin is formed (cluster $(b)c$), two recurrent nonnative states are observed, both of which contain contacts that must be broken prior to subsequent folding steps (Fig 5A, left). Namely, the N-terminal beta strand (residues

5-10) forms nonnative contacts with the C-terminal hairpin by docking with either residues 40–50 (nonnative state 1) or 55–60 (nonnative state 2) in an anti-parallel, rather than parallel orientation, as in the native state. Starting from nonnative state 2, the protein can proceed folding along pathway 1 via closure of the N-terminal beta hairpin. But at this point (cluster $a(b)c$), the N and C termini are still docked in an anti-parallel orientation (Fig 5A, middle), and must separate prior to re-docking in a native parallel orientation (as in the fully-folded cluster $a(b)cd$). In contrast, neither of the nonnative states observed in cluster $(b)c$ are compatible with folding via pathway 2, which requires that the incorrectly-paired termini separate and re-dock correctly prior to the N-terminal hairpin's closure. If this occurs, then the protein enters state $(b)cd$, at which point a different set of nonnative contacts may form, whereby the N-terminal strand incorrectly pairs with the central helix (residues $\sim$35–40, Fig 5A right). But these nonnative contacts are less stable, and observed only $\sim$50% of the time in equilibrium simulations.

## Determining validity of conditions I and II

For those clusters which show nonnative contacts that interfere with folding, we must determine whether conditions I or II applies. If so, then we expect to be able to accurately predict folding times using our method. To determine whether nonnative contacts observed in equilibrium simulations form quickly relative to the timescale of transition between clusters (i.e. condition I is satisfied), we run refolding simulations starting from snapshots assigned to each intermediate with no nonnative contacts initially present (Fig 5B, see also Materials and methods). We find that condition I indeed holds for cluster $(b)c$ (Fig 5B, left). But on the other hand, in the case of cluster $a(b)c$, the observed nonnatives form much more *slowly* than native substructure $d$, which is the next (and final) folding step (Fig 5B, middle). Thus this cluster violates condition I. Moreover, these nonnative contacts seem to rapidly preform while the protein is still in the previous cluster $(b)c$ (Fig 5A). Thus, this cluster also violates condition II, which requires that the system start in the hub state from which productive transitions to the next cluster, $a(b)cd$ can occur. Given that both conditions are violated, we cannot accurately predict the $a(b)c$ to $a(b)cd$ transition time when simulations are initialized from a fully unfolded ensemble (S5A Fig). However, if we instead initialize simulations from $a(b)c$ snapshots in which no nonnatives are present, we artificially ensure that this cluster satisfies condition II, and can now accurately predict the folding time (S5B Fig). Finally we consider cluster $(b)cd$. Although the nonnative contacts observed in this cluster form slowly (Fig 5B, right), in violation of condition I, we note that these nonnatives are absent in previous folding steps. Thus, during *ab initio* folding trajectories, the protein starts in the $(b)cd$ cluster's hub, indicating that condition II applies.

## Computing and verifying folding rates for protein G

We now test whether our method can correctly predict folding times for transitions involving clusters which satisfy conditions I or II. To accomplish this, we incorporate our inferred cluster free energies and extrapolated unfolding rates into the detailed-balance relationship (Eq (3)) to compute the inverse mean-first passage time (MFPT) for each folding transition (Fig 6). This quantity is equivalent to the exponential folding rate (Eq (2)) for clusters that satisfy condition I. We then compare these predictions with observed transition times obtained from serial refolding simulations (See Materials and methods). We find that, for all transitions which satisfy condition I or II, the predicted and observed inverse MFPTs closely agree at physiological temperatures. Where discrepancies are observed, they are typically smaller than an order of magnitude and may result from 1.) Transient misclassification events, which

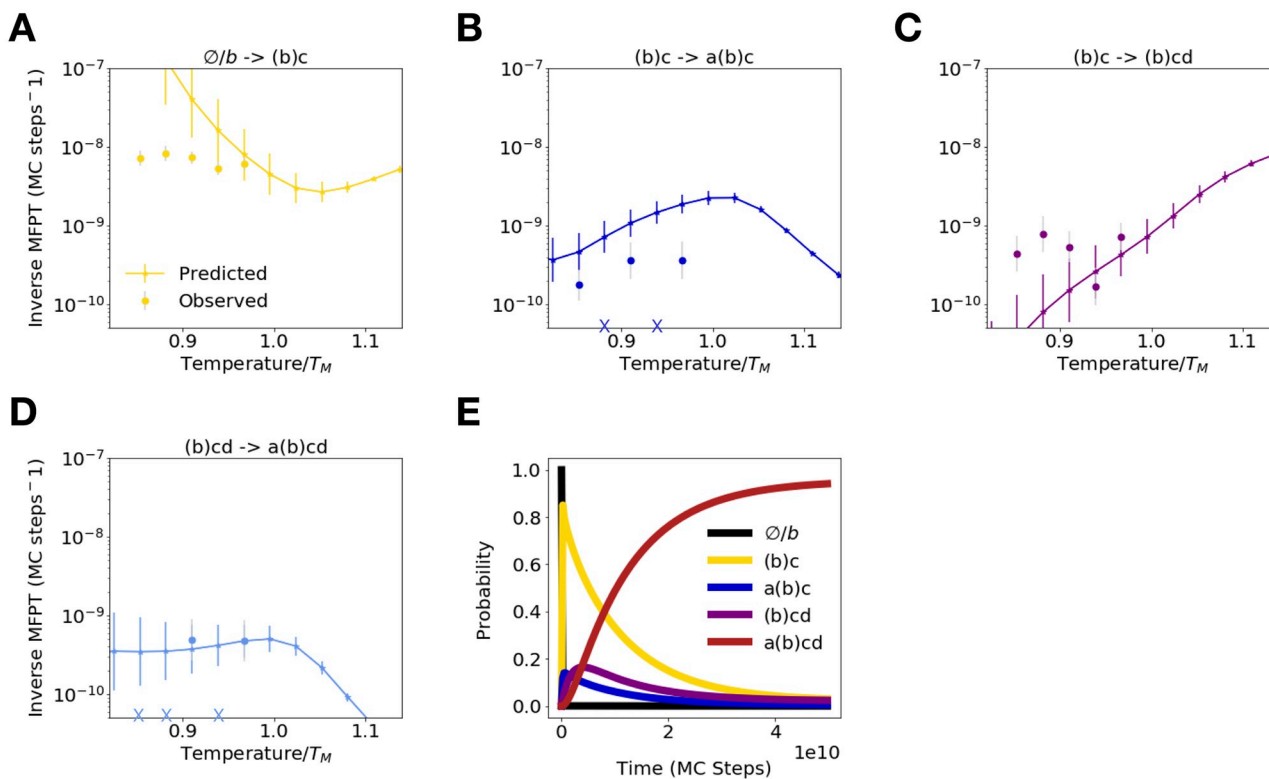

**Fig 6. DBFOLD can accurately predict protein G folding transition times.** (A)-(D) Predicted (markers with errorbars connected by lines), alongside observed inverse mean-first passage times for folding from serial refolding simulations (disconnected round markers), are shown as a function of simulation temperature for transitions between clusters $\emptyset/b$ and $(b)c$, $(b)c$ and $a(b)c$, $(b)c$ and $(b)cd$, and $(b)cd$ and $a(b)cd$, respectively. Error bars represent the standard deviation of bootstrapped error distributions (see Materials and methods). X symbols on x-axis indicate that no folding transition was observed at that temperature. (E) Solution to master equation (Eq (1)) for probability of occupying each cluster as a function of time at a simulation temperature of $T = 0.91\ T_M$.

artificially skew the observed inverse MFPT towards higher values. These are observed particularly often for the $(b)c \rightarrow (b)cd$ transition. 2.) Error in inferring folding rates from refolding trajectories for rare folding events whose inverse MFPT is much less than $10^{-9}$ MC steps. 3.) Imperfect convergence of equilibrium simulations, which may bias our free energy estimations. This latter issue particularly affects the calculation of the $\emptyset/b - >b(c)$ (C-terminal hairpin) folding rate at temperatures below $T \approx 0.9\ T_M$, as the unfolded cluster $\emptyset/b$ is highly unstable at these temperatures and is thus rarely populated, leading to significant error. Despite this uncertainty, we nevertheless observe qualitatively clear anti-Arrhenius behavior with temperature for this transition, which has been previously described theoretically and experimentally for the folding of a simple hairpin without interference from non-native contacts [40–42]. For all other transitions, we observe that as temperature is decreased below the melting temperature, folding rates *decrease* owing to increased stabilization of non-native contacts. In the case of the $(b)cd \rightarrow a(b)cd$ transition, this decrease is more modest, owing to the fact that nonnative traps stabilizing cluster $(b)cd$ are relatively shallow (Fig 5A, right).

In order to approximately track how the populations of different intermediates evolve with time according to our model, we incorporate our extrapolated unfolding and predicted folding rates into the master equation (Eq (1), and numerically solve for the probabilities of occupying the various observed clusters, $P_t(C_i)$, as a function of time, assuming the system starts in the cluster containing the fully unfolded state $\emptyset/b$ (Fig 6E). We omit transitions between $a(b)c$ and

$a(b)cd$ because our method is not able to accurately predict this MFPT, which satisfies neither conditions I nor II. But this transition is never observed in our serial folding simulations, suggesting it is likely very slow compared to all other transitions (S5 Fig). We note that, although we are able to predict inverse MFPTs from $(b)cd$ to $a(b)cd$ (due to condition II being satisfied), this transition will technically show multi-exponential behavior, which we neglect when we treat it as a singe-exponential process in the master equation. Solving the equation, we observe that following rapid folding of the C-terminal beta hairpin (cluster $(b)c$), the majority of the flux rapidly proceeds into pathway 1, entering cluster $a(b)c$. But this pathway represents a trap, owing to the extremely slow (approximated as 0) rate of transition from $a(b)c$ to the fully folded state. Thus, any flux that enters this pathway must backtrack to $(b)c$ before folding can proceed through the productive pathway 2. This requires the separation of the N and C-terminal beta strands, which tend to pair up in a nonnative, anti-parallel registration in clusters $(b)c$ and $a(b)c$. Once equilibrium is reached (around $4*10^{10}$ MC steps), roughly 95% of the population is fully folded at this simulation temperature of $T = 0.91$ $T_M$, although qualitatively similar behavior is observed at other temperatures (S6 Fig).

We further note that we have coarse-grained out the folding of the central helix (substructure $b$) through our clustering method, so as to reduce the complexity of our model and improve fitting quality for unfolding rates. Thus, although we know that helical contacts are capable of dynamically forming and breaking at each stage in the folding process, we cannot quantify the fraction of the time the helix will be folded at each step using this coarse description involving clusters. However, from our PMF at $T = 0.91$ $T_M$ (Fig 3C), we observe that for every cluster beginning with $(b)c$, the state with the helix formed (ex. $bc$) is thermodynamically favored over its counterpart with the helix disrupted (e.g. $c$). This suggests the helix will typically fold early in the folding process, roughly contemporaneously with the folding of substructure $c$, consistent with previous work [32]

## Application of DBFOLD to proteins with complex folding pathways

In the previous sections, we have verified that DBFOLD can accurately predict folding pathways and Monte-Carlo rates for protein G, a small protein for which these predictions can readily be verified via direct folding simulation. But for larger, more complex proteins, DBFOLD can compute folding pathways at a significantly reduced computational cost relative to direct simulation, which would require unreasonably long simulation times. We quantify this computational speedup for various proteins simulated here and in reference [17] by tallying the total time that was required to run equilibrium simulations to convergence, and to collect sufficient high-temperature unfolding statistics to allow for extrapolation prior to use of DBFOLD (Fig 7, rightmost column). These times range from 2 to $5 \cdot 10^9$ MC sweeps, which corresponds to wall times on the order of 10 CPU days on a 4x AMD Opteron 6376 2.3 GHz processor. For each protein, we compared this time to the the average time required for a direct simulation to undergo the rate-limiting folding step in the context of the MCPU potential and moveset (Fig 7, second column to right). These times, which were estimated directly from the predicted rates obtained from DBFOLD, range from $10^6$ MC sweeps for the HEMK N-terminal domain ($\sim 12$ hours of wall time) to $\sim 10^{17}$ MC sweeps for MarR ($\sim 2$ million years of wall time). We note that this $\sim 11$ order-of-magnitude range in predicted folding times is comparable to the experimentally measured range from sub-microseconds to thousands of seconds [40].

From these values, we see that DBFOLD significantly reduces the computational costs required to compute folding properties for the proteins MarR, FabG, and CMK–the latter two of which are larger than 200 AAs in size. In contrast, for the 159 AA protein DHFR, the cost to

| Protein | | Size (AAs) | Temperature at which folding rate computed | Estimated folding time via direct simulation (MC sweeps) | Total simulation duration for *dbfold* (MC sweeps) |
|---|---|---|---|---|---|
| **Protein G** | | 61 | $0.90\ T_M$ | $2 \cdot 10^8$ | $5 \cdot 10^9$ |
| **HEMK NTD** | | 74 | $0.92\ T_M$ | $1 \cdot 10^6$ | $3 \cdot 10^9$ |
| **MarR** | | 144 | $0.51\ T_M$ | $2 \cdot 10^{17}$ | $2 \cdot 10^9$ |
| **DHFR** | | 159 | $0.89\ T_M$ | $3 \cdot 10^9$ | $2 \cdot 10^9$ |
| **CMK** | | 227 | $0.92\ T_M$ | $4 \cdot 10^{16}$ | $2 \cdot 10^9$ |
| **FabG** | | 244 | $0.91\ T_M$ | $5 \cdot 10^{15}$ | $2 \cdot 10^9$ |

**Fig 7. DBFOLD can compute folding pathways for complex proteins at significantly reduced computational cost relative to direct folding simulation.** For each protein, we report the average time in Monte Carlo sweeps (Monte Carlo steps divided by protein size), as predicted by DBFOLD, that would be required for the protein to undergo the rate-limiting folding step within a single direct folding simulation in the MCPU potential at the indicated temperature, normalized by the respective protein's melting temperature $T_M$ (fourth column). This is compared with the total duration of simulations (time per processor multiplied by number of processors) that were run prior to using the DBFOLD algorithm, including both the time for equilibrium simluations to reach convergence and the total duration of high-temperature unfolding simulations. Simulations for Protein G were run in this work (Figs 3–6) while simulations for the other indicated proteins were run in reference [17]. We note that for MarR, simulations were run for the monomeric species, which is relatively unstable in the absence of a dimerization partner, hence a lower simulation temperature was used.

use DBFOLD is comparable to the direct simulation cost, whereas for the sub-100 AA HEMK N-terminal domain and protein G, DBFOLD is significantly *less* efficient than direct folding simulation. This is because these later two proteins' fast folding rates are readily accessible to direct simulation. Indeed in reference [17], we confirm that HEMK NTD folding events are observable within the timescale of a short unfolding/refolding trajectory, as is the case here for protein G. Interestingly, we note that the three proteins for which DBFOLD confers a substantial computational advantage are the same ones that were predicted in reference [17] to benefit from co-translational folding. Namely, these proteins were found to fold very slowly due to nonnative contacts involving C-terminal residues, which can be circumvented by commencing folding at intermediate chain lengths. Thus we observe that DBFOLD particularly facilitates these computations for proteins which undergo slow folding due to deep nonnative traps.

## Discussion

### Summary

We have developed a novel computational method, DBFOLD, that allows for the prediction of protein folding pathways and relative rates of different transitions while accounting for the effects of off-pathway non-native contacts. Rather than directly simulating folding, we show that one can use a combination of equilibrium simulations with enhanced sampling and *unfolding* simulations to compute folding rates using the principle of detailed balance. Our method builds on previous studies that made use of high-temperature unfolding simulations to shed light on protein folding pathways [43, 44]. However, these simulations by themselves cannot be used to obtain rates of folding, nor can they elucidate the role of nonnative contacts, which may stabilize folding intermediates at physiologically-relevant temperatures but be absent at high temperatures at which unfolding simulations are run.

Our work overcomes this limitation by making use of replica-exchange simulations to model intermediate states under physiological conditions. We note, however, that our approach for simulating kinetically-complex proteins differs from previous efforts to extract kinetic information from replica-exchange simulations [45–48]. While these previous methods can accurately compute transition rates for small proteins that rapidly transition between conformers, larger proteins rarely undergo such transitions within simulation timescales in the absence of a biasing potential. Although methods have been developed to obtain rates from biased simulations such as our equilibrium simulations [49], these techniques require knowledge of a functional form for how bias affects transition rates, which may not be known for complex systems. Our method is the first, to our knowledge, which combines biased replica-exchange simulations with high-temperature unfolding simulations to overcome the difficulty in accurately computing transition rates for large proteins. Moreover, while we have chosen to bias our equilibrium simulations along one specific order parameter–namely native contacts–our method does not require us to have chosen an optimal order parameter along which slow transitions occur, as is the case with Metadynamics-based methods [50, 51]. Rather, the purpose of our biasing potential is to promote sampling of distinct *minima* along the folding landscape, and not necessarily the transition saddle points. The free energy barriers associated with these transitions are instead obtained via extrapolation from high-temperature unfolding simulations. Thus, we expect that biasing along alternative order parameters, such as root-mean squared deviation (RMSD) and, radius of gyration ($R_G$), will yield similar results, and believe it would be interesting to compare their efficiencies in future work.

### Validation on protein G

To validate our method, we demonstrate that it can be used to predict folding pathways and Monte-Carlo rates for the well-studied Streptococcal protein G. We find that, for all folding transitions which satisfy certain requirements, the predicted rates agree closely with observed rates, which for this relatively small protein, can be directly computed via *ab initio* simulation. Moreover, these predicted rates paint a picture of complex folding kinetics involving multiple intermediates and off-pathway nonnative states (Fig 6E), consistent with experimental findings that the protein cannot be described as a simple two or three-state folder [36]. Our model further reproduces a number of atomistic-level findings regarding protein G's folding pathway. For example, we observe that the C-terminal hairpin is the first structural element to fold, consistent with a number of previous experimental and computational works [20, 31, 32]. Furthermore we find that pathway 2, in which the N and C-termini dock together immediately

after the C-terminal hairpin forms, represents the dominant pathway through which productive folding occurs. This is consistent with previous work using a native-centric potential [20].

## Use of method to investigate folding of complex proteins

In addition to cross-validating DBFOLD's predictions against direct folding simulations of the fast-folding protein G, we have shown that this method significantly reduces the computational effort required to predict folding pathways and Monte-Carlo rates for larger proteins whose folding is significantly slowed by nonnative contacts. The effects of nonnative contacts may have been significantly underestimated by previous computational studies of protein folding, many of which rely on native-centric (Gö) potentials. These simplified potentials allow complete folding trajectories to be simulated in a reasonable amount of computation time, and may provide a valid description of folding pathways for proteins that are minimally frustrated [52]. However, many proteins, especially larger ones, may suffer from significant nonnative trapping, and their folding thus cannot be accurately described using native-centric potential functions. In contrast to many existing techniques, DBFOLD can generate detailed atomistic predictions of nonnative states and account for their effect on folding times. Thus, the method may be used to shed light on myriad cellular processes where these states play a crucial role including co-translational folding [17], the role of chaperones, and non-native oligomerization or aggregation.

A number of considerations may be relevant when DBFOLD is applied to larger proteins. On the one hand, for large proteins, we expect that nonnative contacts will often form rapidly compared to native contacts, owing to the plethora of nonnative states that are possible alongside the fact that nonnative contacts are often short-range [53, 54]. Thus, transitions between coarse states–each of which consists of all nonnative microstates consistent with a given native topology–are likely to show Markovian behavior (condition I), thus ensuring that detailed balance can accurately be used to infer folding rates between coarse states. But on the other hand, one difficulty with larger proteins may involve extrapolation of unfolding rates, particularly if these proteins are highly kinetically stable and unfold in multiple steps. Under these conditions, unfolding may not occur within a reasonable simulation timescale unless the temperature is increased significantly above the melting temperature. At such high temperatures, the native protein is far from equilibrium and experiences strong unfolding forces that may prevent it from equilibrating at intermediate states. Potential indications of this issue may include non-Arrhenius behavior for transitions beyond the first unfolding event, as well as significant structural differences between intermediates of a given topological configuration that are observed in unfolding simulations and those that are observed in replica simulations at the same temperature. If such indicators are observed, then it may be possible to obtain a better estimate for intermediate unfolding rates by initializing these simulations from the respective intermediates observed in replica simulations, which have equilibrated within their respective free energy basins. This approach was used for the proteins FabG and CMK in reference [17].

A second consideration with running unfolding simulations at temperatures significantly higher than the melting temperature is that the protein may unfold via different sets of topological intermediates at high temperatures than at physiological temperatures. This may hinder the computation and extrapolation of unfolding rates involving physiologically-relevant intermediates. A likely indicator of this is the observation that low-free energy intermediates seen in PMF plots below the melting temperature (e.g. Fig 2C, left) do not correspond to the intermediates via which the protein unfolds at high temperature. This issue may likewise be resolved by initializing unfolding simulations from intermediates that are observed at physiologically-relevant temperatures.

A final consideration for larger proteins is that equilibrium simulations must typically be run for longer in order to achieve convergence. We have observed that we can achieve reasonable convergence for proteins as large as $\sim$300 amino acids within a few weeks of computation time. We suspect that this may represent an approximate upper bound on the size of proteins that can be simulated to convergence within a reasonable timeframe using MCPU, but we have yet to explore this issue systematically.

## Use of DBFOLD to generate experimentally-testable predictions

The atomistic description of folding provided by DBFOLD can be used to generate novel experimentally-testable predictions, including the role of nonnative contacts. In the case of protein G, we observe that the N-terminal beta strand (residues 5–10) frequently docks with the C-terminal hairpin in a non-native, anti-parallel fashion early during folding (Fig 4A). These nonnative contacts persist for a significant amount of time, sometimes causing the protein to enter the non-productive pathway 1 whereby both hairpins close while misaligned. This leads to an off-pathway kinetic trap. Although previous work has also identified nonnative states involving these hairpins docked in an anti-parallel orientation [38], our work provides a quantitative estimate of the effect of these nonnative contacts on folding kinetics. Namely, our finding that a significant fraction of the population adopts long-lived states in which the hairpins misalign in an antiparallel fashion (Fig 6E) suggests these nonnative contacts should be observable by FRET. Our method can also be applied to predict how sequences changes affect a protein's folding pathway. For instance, previous work suggests that in protein G's close structure homolog, protein L, residues in the N-terminal hairpin show the highest $\phi$-values [55]. This is in contrast to protein G, where the highest $\phi$-values are found in the C-terminal residues [31]. Potentially consistent with this, we find that DBFOLD predicts folding intermediates involving the N-terminal hairpin to be $\sim$5–10 $k_BT$ more stable in protein L than in protein G at physiological temperatures (S11 Fig). But a more quantitative comparison with experimental $\phi$-values would require computation of folding stabilities and transition state barriers for all the relevant mutants within our potential, either using our method or less costly, approximate methods [20]. Our current simulations also predict that the ratio of the N-to-C terminal hairpin folding rates is higher for protein L than for protein G (S12 Fig), although the simulations do not predict a complete shift in the folding flux towards the N-terminal pathway. But this incomplete shift does not necessarily contradict experimental $\phi$-values, which sometimes fail to unambiguously predict the structure of the transition state ensemble. For instance, studies using $\psi$-value analysis paint a different picture whereby all four beta strands are formed in the transition states for both homologs [38].

In addition to proteins G and L, we expect that DBFOLD may accurately predict sequence effects on the folding of proteins for which the potential has successfully predicted mutational changes before. These include *E. Coli* DHFR [56] and human $\gamma D$-crystallin [57]. Finally, our method has been previously applied to generate atomistic-level predictions of how complex, misfolding-prone proteins may begin folding co-translationally [17]. These predictions can be readily tested by purifying and biophysically characterizing protein fragments of different lengths.

When DBFOLD is used alongside Monte-Carlo simulation algorithms such as MCPU, it can predict *relative* rates of different folding transitions for a given protein, but not absolute rates in measurable units, as the predicted rates are all in Monte-Carlo units. Knowledge of these relative rates is nevertheless useful–for instance, it allows for rate-limiting folding step(s) to be identified. Likewise, kinetic models can be generated that predict the relative populations of different folding intermediates over time (e.g. Fig 6E), even if the absolute timescale over

which this evolution occurs is not known. In order to convert Monte-Carlo rates to experimentally-measurable units, it will be necessary to benchmark the MC simulations against experimentally measured folding times. Alternatively, DBFOLD could in principle be utilized with molecular dynamics (MD) simulations, which have the obvious advantage of predicting folding rates in absolute, measurable units. We expect the method to work well with MD so long as the requirements presented here are satisfied. In particular, it will be important to verify that unfolding rates can still be well-fit to the Arrhenius equation. The presence of explicit solvent may introduce a temperature-dependence to the unfolding-rate prefactors and activation energies, in which case an alternative model should be used for rate extrapolation [58]. However, it is worth noting that the small simulation timesteps used in MD render these simulations much slower to run than MC simulations. Even with the use of replica exchange and enhanced sampling, the timescales required to achieve convergence in MD simulations may significantly exceed accessible computation times, particularly for large proteins [24, 59, 60].

## Materials and methods

### Atomistic Monte-Carlo simulations

In principle, our method can be applied in conjunction with any protein molecular dynamics or Monte Carlo simulation software so long as detailed balance is obeyed. Here, we utilize an atomistic Monte Carlo (MC) simulation package, MCPU, described in previous works [17, 61–63]. This package, whose latest version is available online as part of DBFOLD, uses a knowledge-based potential to rapidly compute energies while accounting for both native and non-native interactions. We model all backbone and sidechain atoms with the exception of hydrogen, and assign to each configuration an energy contains terms accounting for contacts between atoms, hydrogen bonding, relative orientation of aromatic residues, as well as local backbone and side chain torsion angle strain. Our MC moveset allows for both sidechain and backbone rotations, as well as "local moves" which modify the dihedral angles of only three consecutive residues while keeping the rest of the backbone intact. In order to ensure detailed balance is satisfied, a proposed move from a configuration with atomic coordinates $\mathbf{x_n}$ to one with coordinates $\mathbf{x_m}$ is accepted with probability given by the Metropolis-Hastings Criterion:

$$P_{n \rightarrow m} = \min \left( 1, \frac{J(\mathbf{x_m})}{J(\mathbf{x_n})} \exp \left\{ -(E(\mathbf{x_m}) - E(\mathbf{x_n}))/k_B T \right\} \right) \tag{5}$$

Where $E(\mathbf{x_n})$ and $E(\mathbf{x_m})$ are the energies of the respective configurations while $J(\mathbf{x_n})$ and $J(\mathbf{x_m})$ are Jacobian determinants that account for changes in the size of phase space owing to local moves–for details see [64, 65]

To compute a protein's thermodynamic properties, we deploy enhanced sampling techniques in order to aid convergence to equilibrium. First, for each trajectory, we add to the energy function a harmonic biasing term that encourages the simulation to explore configurations with a number of native contacts in the vicinity of some setpoint $S$. Namely, a configuration with unmodified energy $E(\mathbf{x_n})^0$ (computed as described above) and number of native contacts $N(\mathbf{x_n})$ will be assigned a modified energy given by

$$E(\mathbf{x_n}) = E(\mathbf{x_n})^0 + \frac{1}{2} k_{\text{bias}}(N(\mathbf{x_n}) - S)^2 \tag{6}$$

Equilibrium simulations are run at a range of temperatures and setpoints as described below. To further aid convergence, we implement replica exchange in which pairs of simulations with adjacent setpoints or temperatures periodically attempt to swap configurations. Suppose the two trajectories that attempt an exchange have setpoints $S$ and $S'$ and

temperatures $T$ and $T'$, respectively. If these two trajectories initially populate configurations $\mathbf{x_n}$ and $\mathbf{x_m}$ with (unmodified) energies $E(\mathbf{x_n})^0$ and $E(\mathbf{x_m})^0$, and numbers of native contacts $N(\mathbf{x_n})$ and $N(\mathbf{x_m})$ respectively, the probability that the exchange is accepted is given by

$$
P(\mathbf{x_n} \leftrightarrow \mathbf{x_m}| \quad (T,S),(T',S')) = \min\left(1, \quad \exp\left[\left(\frac{1}{T'} - \frac{1}{T}\right)\left(E(\mathbf{x_m})^0 - E(\mathbf{x_n})^0\right)\right.\right.
$$

$$
\left.\left. -k_{\text{bias}}((N(\mathbf{x_m})-S)^2 - (N(\mathbf{x_n})-S)^2 + (N(\mathbf{x_n})-S')^2 - (N(\mathbf{x_m})-S')^2)\right]\right)
$$

(7)

In our simulations, we implement exchanges every 500,000 MC steps, at which 75 pairs of cores with adjacent setpoints or temperatures are randomly chosen to attempt an exchange.

All simulations except refolding simulations are initialized from a relaxed structure, which is generated from a starting crystal structure (PDB ID 1igd for protein G) by first running a simulated annealing protocol whereby the protein is subjected to gradually decreasing temperatures from T = 0.45 to T = 0.1, and allowed to equilbrate for 2 million MC steps at each temperature without umbrella biasing. Next we run an initial set of replica exchange simulations starting from the annealed structure in the previous steps, with multiple umbrella setpoints and temperatures as low as T = 0.2. Together, these two steps increase the likelihood that the protein will undergo small conformational changes needed to reach the energy minimum in this potential. The lowest energy structure after the second step is defined as the equilibrated native structure, which is used to initialize the subsequent simulations:

1. Equilibrium simulations are run using a grid of setpoints ranging from 0 to the total number of native contacts (42 for protein G) rounded to the nearest ten in increments of $\Delta S = 10$, and simulation temperatures typically ranging from from $T = 0.4$ to $T = 1$ in increments of $\Delta T = 0.025$ in simulation temperature units. The number of native contacts is computed by counting the number of alpha carbon (CA) pairs whose distance within the equilibrated native structure is less than some cutoff (typically 6 Angstroms for predominantly beta-sheet proteins such as protein G and 8 Angstroms for helical proteins such as MarR in which residues often interact via their side-chains, thus resulting in larger CA distances) and whose separation in sequence is at least 8 residues (to include only tertiary contacts). A value of $k_{\text{bias}} = 0.02$ is used in simulation energy units. Simulations are run until convergence is reached, as assessed using the methods described in the next subsection.

2. High temperature unfolding simulations that implement neither umbrella biasing (i.e. with $k_{\text{bias}} = 0$) nor replica exchange are run at a range of temperatures above the melting temperature for 100 million MC steps.

We additionally run refolding simulations, which are initialized from various structural intermediates, as described in the main text. These simulations again deploy neither umbrella bias nor replica exchange.

## Assessing convergence

To assess the convergence of our equilibrium simulations, we compute the simulation energy, averaged over a sliding window, as a function of MC step for each umbrella at various temperatures (S2A–S2C Fig). For protein G, we find that by 1 billion Monte Carlo steps, these average energies stop changing, suggesting convergence. This is further supported by examining a more global metric of convergence–namely for each trajectory with setpoint $s$, we compute the deviation between $s$ and the number of native contacts at each MC step. We then compute the root-mean-square of this deviation, averaged over all trajectories with setpoint $s$ across all

temperatures and over a sliding window of 50 million MC steps. (S2D Fig). Based on these metrics, we compute equilibrium properties using the last 150 million MC steps, but our results do not significantly change if we slightly vary the MC steps used (S2E and S2H Fig)–an additional indicator of convergence.

## Substructure analysis

To identify native substructures for a given protein, we produce a contact map of the equilibrated native structure by identifying alpha-carbon (CA) pairs that are separated by at least 8 amino acids (for MarR and DHFR) or 3 amino acids (for protein G) in primary sequence and whose spatial separation is within some distance cutoff $d_c$. We set $d_c$ = 6.5 angstroms for the predominantly beta proteins DHFR and protein G, and $d_c$ = 7.8 for the predominantly helical protein MarR. We then define substructures as islands of native contacts comprised of at least $c$ contacts ($c$ = 7 for MarR and protein G, while $c$ = 12 for DHFR) which can be entirely traversed via hops of Manhattan distance no greater than $h$ ($h$ = 3 for MarR while $h$ = 5 for DHFR and protein G) within the contact map. Values for $d_c$, $c$, and $h$ are adjusted for each protein so as to ensure all major structural units are included, while excluding substructures with few contacts which are expected to rapidly form/break, thus avoiding overfitting.

To determine whether substructure $s$ is folded within a snapshot, we compute the average spatial distance $\langle d \rangle$ between every pair of CAs that form contacts assigned to $s$. We then compute that same average distance in the equilibrated native file $\langle d_0 \rangle$. So long as $\langle d \rangle \leq f \langle d_0 \rangle$ for some factor $f$, then the substructure is deemed formed. For predominantly beta sheet proteins such as protein G, we set $f$ = 1.7, but larger values are used for helical proteins where contacts may involve side chains (and thus larger distances between CAs). We show in S8 Fig that our results for protein G are robust over a reasonable range of values for $f$. We can now assign a snapshot to a topological configuration labeled by the substructures that are formed within that snapshot. We note that in some cases, this algorithm may declare a substructure as formed even if the registration between the interacting residues is slightly shifted or, in the case of beta sheet substructures, if the two strands interact with nearly the correct registration but using a non-native set of hydrogen bond donors/acceptors within the backbone. But we do not expect this to be a serious limitation as such slightly-nonnative conformations are expected to rapidly exchange with the substructure's native conformer.

## Computation of thermodynamic properties from equilibrium simulations

From our equilibrium simulations with enhanced sampling, we can compute relative free energies, and thus equilibrium probabilities of different states using MBAR [23], a statistically optimized method for computing free energies under a desired set of conditions (for example, a specific temperature and no bias). MBAR takes advantage of data sampled at all conditions (i.e. trajectories at all temperatures and biases) to compute equilibrium probability $P_{eq}^{T}(X)$ of observing some observable value $X$ at temperature $T$ (in the absence of umbrella biasing). $X$ may correspond, for instance, to some topological configuration or some number/fraction of native contacts. Given these equilibrium probabilities, we can define a dimensionless potential of mean force (PMF) at temperature $T$ as a function of $X$ as:

$$F_i^T(X) = -\log\left(P_{eq}^T(X)\right) \tag{8}$$

In Fig 4C, we show PMF values for each coarse state computed from the equilibrium probabilities as above. From these probabilities, we can also compute the ensemble average of $X$ over

all observed values, $X_i$, as

$$\langle X \rangle = \sum_i X_i P_{eq}^T(X_i))$$

(9)

Fig 4B shows the mean number of native contacts as a function of temperature, computed in this way.

## Computing and extrapolating unfolding rates

To obtain transition rates from high temperature unfolding simulations, we first assign all observed snapshots to their respective topological configurations. For the larger proteins simulated in reference [17], we applied an additional filtering step to eliminate configurations that are rarely observed, thereby reducing the number of total configurations and parameters in the model. Namely, any snapshot assigned to a topological configuration that encompasses less than some fraction $s$ (typically $s = 1\%$) of all unfolding simulation snapshots is reassigned to either the configuration that came before or after it in the trajectory, depending on which it is most topologically similar to. In the main text, this step was not applied for protein G, where the total number of configurations occupied was small. However, we show in S9 Fig that applying this step does not appreciably change the final folding MFPT predictions. Next, we attempt to reduce instances in which snapshots are misassigned to the wrong topological configuration by training a Hidden Markov Model (HMM) on data from all temperatures using an emission matrix that assumes a 90% probability that a topological configuration is classified correctly, and a 10% probability of incorrect classification, uniformly distributed over all observed incorrect states. Varying this misassignment probability, $m$, does not significantly change results (S10 Fig). This HMM is then used to fit the maximum-likelihood sequence of configurations for each trajectory. Finally, we further reduce the number of parameters via an additional clustering step that groups configurations in fast exchange. To this end we compute a "kinetic distance" $T_K^{i,j}$ between every pair of observed configurations i and j, defined as the average time to transition between them in either direction, given that the system is in one of the two states. We then define two configurations as adjacent if their kinetic distance is less than some threshold, and define clusters as connected components of the resulting adjacency matrix. The value of the adjacency threshold $T_A$, set to 100 million MC steps for protein G, is chosen to lie between any highly separated timescales that are observed, such that configurations within a cluster exchange much faster than configurations in different clusters (S7 Fig). This criterion ensures that clusters obey the criteria outlined in Conditions I and II in the main text.

Having assigned trajectory snapshots into clusters, we now estimate the survival probability, $P_S(C_j, t, T)$ for cluster $C_j$, namely the probability that a trajectory which transitioned into cluster $C_j$ from some other cluster at t = 0 will not yet have made any excursions out of $C_j$ after time t has elapsed at temperature T (as in Fig 5C). We can likewise estimate conditional probability, given that the protein is initially in cluster $C_j$ at temperature T, of transitioning to some other cluster $C_i$ during the inter-snapshot time interval $\Delta t$ (which is typically 500,000 MC steps)

$$P(C_i|C_j, T, \Delta t) = \frac{Z_{C_j \to C_i}^T}{Z_{C_j}^T}$$

(10)

Where $Z_{C_j \to C_i}^T$ is the total number of observed transitions between clusters $C_j$ and $C_i$ at temperature T, and $Z_{C_j}^T$ is the total number of snapshots at temperature T assigned to cluster $C_j$. We

then convert this to a rate

$$k_{j \to i}(T) = \frac{1}{\Delta t} \log \left( \frac{1}{1 - \sum_{l \neq j} P(C_l | C_j, T, \Delta t)} \right) \frac{P(C_i | C_j, T, \Delta t)}{\sum_{l \neq j} P(C_l | C_j, T, \Delta t)} \tag{11}$$

This assumes a continuous-time Markov process, but in case the transition from $C_i$ to $C_j$ is non-Markovian (e.g. if condition II is satisfied, but not condition I), then this equation nonetheless provides a good approximation to the inverse mean-first passage time to transition, so long as $\sum_{l \neq j} P(C_l | C_j, T, \Delta t)$ is small–i.e. transitions during a single MC step are unlikely. For every pair of clusters $C_j$ and $C_i$, we compute $k_{j \to i}(T)$ at all temperatures at which transitions are observed, and the dependence on inverse temperature is fit to the Arrhenius equation (Eq (4)). To obtain an error distribution on the extrapolated unfolding rates due to finite sampling, we perform a bootstrap analysis whereby, at each temperature, N trajectories (where N is the original total number of trajectories at that temperature) from the original set are randomly sampled with replacement. The log unfolding rates for these resampled trajectories are again fit to the Arrhenius equation, and re-extrapolated to lower temperatures using the resulting parameters. This process is repeated 1000 times. We note that in the main text (Fig 3), all unfolding temperatures are included in Arrhenius plots, but during folding rate computation, for each transition we only perform Arrhenius analysis with temperatures that show five or more transition events. Applying this threshold does not significantly change results for protein G, but is a recommended practice in general to reduce noise.

## Identifying nonnative states

We generate nonnative contact maps for each cluster by pooling all snapshots from equilibrium simulations assigned to topological configurations within that cluster. Nonnative contacts are defined as contacts whose Manhattan distance on the contact map is at least 2 from any contact present in the equilibrated native contact map (so as to exclude native contacts that have been slightly register-shifted), with the exception of cluster $(b)c$ where only native contacts are excluded. We then subdivide these contact maps as in reference [17]–briefly, we identify the connected components of the adjacency matrix between different snapshots' nonnative contact maps, where two maps are defined as adjacent if their hamming distance is less than 10. For cluster $(b)c$, two structurally-distinct classes of nonnative states were identified using this method, whereas only one such class was identified for all other clusters. For each class of nonnative states, we then compute averaged nonnative contact maps among all snapshots assigned to that class (Fig 5). We then identify nonnative substructures among all nonnative contacts that are present at least 10% of the time within a given nonnative class using the methods described in the subsection Substructure analysis with values of $c = 7$, $d_c = 6.5$ Angstroms, and $h = 3$. For each cluster $C_i$, we then run refolding simulations, initialized from snapshots assigned to $C_i$ with no more than 2 nonnative contacts, drawn from equilibrium simulations at $T \approx 1.14\ T_m$, and compute the probability, as a function of MC step, that the protein forms at least one nonnative substructure obtain from the nonnative contact maps generated as above for cluster $C_i$ or the probability that the protein has transitioned to a subsequent cluster in the folding pathway (Fig 4B).

## Computing predicted and observed folding rates

To compute the predicted folding rate $k_{i \to j}^T$ between clusters $C_i$ and $C_j$ at temperature $T$, we incorporate equilibrium probabilities of occupying the respective clusters, as well as the unfolding rate $k_{j \to i}^T$ (computed and extrapolated as per the previous section) into Eq (2)). We

obtain

$$k_{i \to j}^T = \frac{\sum_l \mathbb{1}_{S_l \in C_j} P_{eq}^T(S_l)}{\sum_{l'} \mathbb{1}_{S_{l'} \in C_i} P_{eq}^T(S_{l'})} k_{j \to i} \tag{12}$$

Where the sum is over all observed coarse states, $\mathbb{1}_{S_l \in C_j}$ has value 1 if coarse state $S_l$ is assigned to cluster $C_j$ and 0 otherwise, and $P_{eq}^T(S_l)$ is the equilibrium probability of occupying coarse state $S_l$ computed as per the subsection "Computation of thermodynamic properties from equilibrium simulations" (and likewise for cluster $C_i$). As discussed previously, this rate corresponds to an inverse mean-first passage time for transitions that do not show Markovian behavior. We obtain an error distribution on this folding rate by incorporating the value of $k_{j \to i}$ obtained from each bootstrap iteration, as described in the subsection Computing and extrapolating unfolding rates, into the above equation. Error bars in plots of $k_{i \to j}^T$ as a function of temperature represent the standard deviation of this bootstrapped distribution, omitting any bootstrap iterations in which the respective unfolding transition $k_{j \to i}$ does not occur. We note that error bars appear symmetric on a log scale because the boostrap analysis is performed entirely in log space. We further note that this error does not account for uncertainty in the PMF calculation.

In order to directly compute refolding rates, we run a set of refolding simulations in which protein G is initialized from snapshots assigned the fully unfolded configuration $\emptyset$ at high temperatures (under the restriction that no more than two nonnative contacts are initially present). At each of five physiolgoically-reasonable temperatures, 120 refolding trajectories are run, starting from snapshots drawn randomly (with replacement) from among those which satisfy the required criteria. These trajectories are run for 100 million MC steps, at which point snapshots that successfully transitioned into cluster $(b)c$ are used to initialize 120 new simulations. This process is repeated serially for each step in the two pathways shown in Fig 5. To compute observed folding rates from these forward folding simulations, we use the HMM method described in the subsection Computing and extrapolating unfolding rates to reduce the frequency with which simulation snapshots are misassigned to the wrong topological configuration. We then assign each snapshot to whichever cluster contains the topological configuration to which that snapshot was assigned (where clusters are determined from the unfolding simulations). Finally, Eqs (10) and (11) are used to compute the observed folding rate from cluster $C_i$ to $C_j$ at each temperature at which folding simulations are run. As with the predicted folding rates, a bootstrap analysis, in which $N$ refolding trajectories at each temperature are resampled 1000 times with replacement, is used to obtain an error distribution on these observed refolding rates.

## Supporting information

**S1 Text. Details on conditions for applicability of method, justification of Arrhenius kinetics.**
(PDF)

**S1 Fig. Schematic for model of transition between coarse states.** $S_i$ and $S_j$ represent coarse states, each of which is composed of the set of all microstates with topological configuration $i$ or $j$, respectively. $S_i$ can transition to $S_j$ if the two differ by the formation/breaking of one substructre. $S_i^h \subseteq S_i$ represents the subset of microstates within $S_i$ from which transitions to states in $S_j$ are possible (i.e. the hub state). All other microstates $s_i^n \in S_i$ cannot transition to $S_j$ owing to nonnative contacts that interfere with folding/unfolding of the required native substructure. Transitions between states are indicated by double arrows and rates denoted by the variable $\lambda$.

For further details, see S1 Text.
(TIF)

**S2 Fig. Assessing convergence.** (A)-(C) Energy is plotted as a function of Monte Carlo step, averaged over a sliding window of 50 million Monte Carlo steps, at three temperatures shown above plots, where $T_M$ is the melting temperature. At each temperature, trajectories for each umbrella setpoint are shown (see legend in left-most panel). At around 1 billion MC steps, these sliding-window averaged energies cease changing substantially, indicative of convergence. (D) For each setpoint, we compute the root-mean-squared deviation between the number of native contacts and the respective setpoint value averaged over all temperatures and over a sliding window of 50 million MC steps. This quantity is plotted as a function of MC step, with different colors corresponding to setpoints as in panel (A). This RMSD stops varying after about 1 billion MC steps, indicating convergence. (E)-(G) Potentials of mean force (PMF) as a function of topological configuration are plotted at a simulation temperature of $T = 0.91\ T_M$ (as in main text Fig 3C), but we now vary the window of simulation timesteps that is used to compute the PMFs, namely we use either steps 900 million through 1.05 billion (E), 975 million through 1.125 billion (F), or 1.05 billion through 1.2 billion (G). Quantitative similarity between these PMFs indicates that thermodynamic quantities are well converged. (H) Thermally averaged fraction of native contacts as a function of simulation temperature (as in main text Fig 3B) using the same MC timestep windows as in panels (E) -(G) (see legend). These curves are nearly superimposable, indicating that thermodynamic quantities are well converged.
(TIF)

**S3 Fig. Dwell-time distributions for clusters at different temperatures.** (A)-(E) Survival probability as a function of Monte Carlo time for each cluster (dots) alongside exponential fits (solid lines) during unfolding simulations, as in main text Fig 3C, at simulation temperatures indicated above the respective panels. Panel (C) shows data at a simulation temperature of 1.08 $T_M$, as in the main text. We note that survival curves for a cluster $C$ at temperature $T$ can only be meaningfully computed over a MC time interval $\tau = \tau_{\text{traj}} - \max_i(\tau_i^0)$, where $\tau_{\text{traj}} = 10^8$ MC steps is the total duration of each trajectory, $\tau_i^0$ is the first MC timestep at which trajectory $i$ (at temperature $T$) samples cluster $C$, and thus $\max_i(\tau_i^0)$ is the latest MC timepoint at which any trajectory at temperature $T$ reaches cluster $C$ for the first time. For all clusters except $abcd$ (at which all trajectories are initialized), $\tau$ is shorter than the total simulation trajectory, hence these clusters' survival curves are truncated. Most curves show good fits to a single-exponential decay. The fits for cluster $(b)cd$ are worse because the $(b)cd$ to $(b)c$ transition satisfies condition II, but not condition I, implying multi-exponential kinetics. We note that these exponential fits are merely used to assess whether clusters satisfy condition I, and not to infer transition rates–these rates are instead inferred using Eqs (10) and (11) in the main text.
(TIF)

**S4 Fig. Nonnative contacts in $\emptyset/b$ cluster do not impede subsequent folding step.** Average nonnative contact map (as in main text Fig 5A) for snapshots assigned to cluster $\emptyset/b$ at temperatures around $T \approx 0.85\ T_M$, alongside an example of such a snapshot with nonnative contacts highlighted. Residues that participate in substructure $c$, the first to form in the folding pathway, are outlined in gray. The predominant nonnative contacts do not impede the impede the formation of substructure $c$–this is further evidenced by the fact that these same nonnative contacts are observed in cluster $(b)c$ (See Fig 4A).
(TIF)

**S5 Fig. DBFOLD fails to accurately predict rates of transitions that satisfy neither conditions I nor II.** (A) Predicted inverse mean-first passage times (MFPTs) to folding (markers with errorbars connected by lines), alongside observed inverse MFPTs from serial refolding simulations (disconnected round markers), are shown as a function of simulation temperature for the transition from clusters $a(b)c$ to $a(b)cd$ (analogous to main text Fig 6). Although the algorithm predicts inverse folding times of approximately $10^{-8}$ MC steps$^{-1}$ (implying that roughly one folding transition should occur per simulation trajectory) at physiologically-reasonable temperatures, in reality no transitions are observed at all at any temperature (as indicated by X's near the x-axis), implying that the true folding rate is likely less than $10^{-10}$ MC steps$^{-1}$. This significant discrepancy occurs because, in the context of serial refolding simulations, neither condition I nor II is satisfied for the $a(b)c$ to $a(b)cd$ transition. (B) Same as (A), but markers now show observed $a(b)c$ to $a(b)cd$ inverse mean first passage times within simulations that are initialized from $a(b)c$ snapshots drawn from temperatures above the melting temperature with no more than two nonnative contacts. This setup artificially ensures that Condition II holds for this transition. In contrast to panel (A), all observed transition rates now lie within a factor of two of the predicted rates, indicating significantly improved predictions.
(TIF)

**S6 Fig. DBFOLD predicts similar folding kinetics for protein G over a range of reasonable temperatures.** (A)—(E) As in main text Fig 6E, we solve the master equation, which incorporates folding and unfolding rates between clusters computed as described in the main text, for the probability of occupying different clusters as a function of time at five different physiologically reasonable temperatures indicated above the respective panels. Panel (C) shows the solution at $T = 0.91\ T_M$ as in the main text. All temperatures show qualitatively similar kinetics, although quantitative details slightly differ between temperatures. In particular, at lower temperatures, relaxation dynamics are slower due to increased stability of nonnative contacts (note the different x-scale in panel (A)), but the fully folded state is also more stable, leading to a higher final equilibrium population of cluster $a(b)cd$.
(TIF)

**S7 Fig. Exchange times between topological configurations show a significant separation of timescales.** The kinetic distance $T_K^{i,j}$–namely the average time to transition between topological configurations $i$ and $j$ in either direction at any temperature conditioned on the fact that the system is in one of the two states –is shown between every pair of topological configurations observed in unfolding simulations. Values written out inside the heatmap are in units of millions of MC steps. This provides a metric for the speed at which topological configurations exchange with each other. We note a significant separation between the fastest timescales of exchange (less than 10 million MC steps) and all slower timescales (greater than 150 million MC steps). Thus, a kinetic threshold of $T_A = 100 * 10^6$ MC steps (see main text Methods section) produces meaningful kinetic clusters.
(TIF)

**S8 Fig. DBFOLD predictions are robust to choice of $f$ value within a reasonable range.** (A)-(D) Predicted folding rate as a function of simulation temperature for each transition that satisfies either condition I or II (as in main text Fig 6) for different values of $f$, which refers to the maximum allowed ratio of the average distance between residues assigned to a substructure in a snapshot divided by that same average distance in the equilibrated file such that the substructure is deemed folded in that snapshot (See main text Materials and methods, subsection "Substructure analysis"). Different linestyles indicate different values of $f$ (as per legend in panel B),

We note that for all transitions, small deviations of $f$ away from the value used in the main text, namely $f = 1.7$, do not significantly change the predicted folding rates, indicating that $f = 1.7$ is a reasonable value. However, larger deviations in $f$ produce more significant changes to these predictions. If our threshold for declaring a substructure folded is too strict ($f \sim \leq 1.5$), then the algorithm tends to overpredict unfolding events, whereas too lenient a threshold ($f \sim \geq 2$) causes the algorithm to underpredict such events. We note that for the extreme values of $f$, the kinetic clusters slightly change. Namely when $f = 1.5$ all clusters except $a(b)cd$ include only the respective topological configuration in which $b$ is unfolded (e.g. $(b)cd$ now only includes $cd$), as $b$ is rarely declared folded under this strict threshold. Conversely, when $f = 2$, clusters $a(b)cd$ and $(b)cd$ include only the respective configuration in which $b$ is folded, as $b$ is rarely declared unfolded under this lenient threshold. Finally we note that higher $f$ values may reasonably be used for predominantly helical proteins where interactions between contacting residues typically involve sidechains, as opposed to backbone hydrogen bonds as in predominantly sheet proteins such as protein G.
(TIF)

**S9 Fig. DBFOLD predictions are robust to choice of $s$ value within a reasonable range.** Predicted folding rate as a function of simulation temperature for each transition that satisfies either condition I or II (as in main text Fig 6) for different values of $s$. Following initial assignment, all topological configurations that encompass less than a fraction $s$ of snapshots over all unfolding simulations are reassigned to the most similar topological configuration that is represented at a fractional prevalence greater than $s$ (see main text Materials and methods, subsection "Computing and extrapolating unfolding rates"). This is useful for larger proteins to reduce the number of unfolding rate parameters. In the main text, a value of $s = 0$ is used (no reassignment), but these panels show that a value of $s = 0.001$ does not drastically change any results. This value reassigns states that are not part of the two predominant folding/unfolding pathways, but leave intact assignments for states belonging to these pathways. Meanwhile, a value of $s = 0.005$ does not drastically change results with the exception of the $(b)c \rightarrow a(b)c$ transition rates. This is because, for this relatively large value of $s$, configuration $a(b)c$, which is observed with frequency less than 0.005 among all snapshots despite belonging to one of the dominant folding/unfolding pathways, gets reassigned. These results therefore indicate that the value of $s$ should always be small enough that topological configurations belonging to the dominant pathways are not reassigned.
(TIF)

**S10 Fig. DBFOLD predictions are robust to choice of $m$ value within a reasonable range.** Predicted folding rate as a function of simulation temperature for each transition that satisfies either condition I or II (as in main text Fig 6) for different values of $m$, the misassignemnt probability in the hidden Markov model used to reduce misassignment of snapshots to incorrect topological configurations (see main text Methods section, subsection Computing and extrapolating unfolding rates). A value of $m = 0.1$ is used in the main text, but these panels show that our results are robust even if $m$ is varied over an order of magnitude.
(TIF)

**S11 Fig. DBFOLD predicts greater stability for N-terminal hairpin in protein L as compared to protein G.** We compute potentials of mean force (PMFs) as a function of topolgoical configuration (as in main text Fig 3C) for protein G (left panels) and its close structural homolog protein L (right panels) at temperatures of $T = 0.88 \, T_M$ (top row) and $T = 0.94 \, T_M$ (bottom row), where $T_M$ is the melting temperature for protein G. Equilibrium simulations for protein L were run for $\sim 1.5$ billion MC steps, then analyzed in an analogous fashion as for protein G

(see Methods in main text), starting from the crystal structure with PDB ID 2ptl. Substructures for protein L are defined nearly identically as for protein G. As shown here, at temperatures below the melting temperature, topological configurations in which the N-terminal hairpin (substructure *a*) is folded but not the C-terminal hairpin (namely configurations *a*, *ab*, and *abd*) are lower in free energy in protein L than they are in protein G. This is consistent with phi-value analysis, which suggests that the N-terminal hairpin in isolation is more stable in the former protein.
(TIF)

**S12 Fig. DBFOLD predicts increased folding flux through the N-terminal pathway in protein L as compared to protein G.** For protein G (red curve with markers and error bars) and protein L (green curve with markers and error bars), we compute the ratio of the rate at which the N-terminal hairpin folds starting from the unfolded state (transition $\emptyset/b - >a(b)$) to the rate at which the C-terminal hairpin folds (transition $\emptyset/b - >(b)c$). Error bars are obtained via bootstrapping is in main text Fig 6. We observe that for both proteins, the N-terminal hairpin's folding is significantly slower than that of the C-terminal hairpin at temperatures below the melting temperature $T_M$. But for protein L, this ratio is higher, indicating increased N-terminal folding as compared to protein G. Thus, although our MCPU potential does not predict a complete change in folding flux towards the N-terminal pathway in protein L, it nevertheless captures a partial shift which is potentially consistent with experimental $\phi$-values. We note that for protein G, it was necessary to initialize simulations from the $a(b)$ cluster in order to compute the N-terminal folding rate. This is because, during simulations initialized from the native state, a very low amount of flux through the N-terminal unfolding pathway was observed, thus precluding the collection of sufficient statistics for Arrhenius fitting.
(TIF)

## Acknowledgments

The computations in this paper were run on the FASRC Cannon cluster supported by the FAS Division of Science Research Computing Group at Harvard University.

## Author Contributions

**Conceptualization:** Amir Bitran, William M. Jacobs, Eugene Shakhnovich.

**Data curation:** Amir Bitran.

**Formal analysis:** Amir Bitran.

**Funding acquisition:** Eugene Shakhnovich.

**Investigation:** Amir Bitran.

**Methodology:** Amir Bitran, William M. Jacobs, Eugene Shakhnovich.

**Project administration:** Eugene Shakhnovich.

**Software:** Amir Bitran.

**Supervision:** William M. Jacobs, Eugene Shakhnovich.

**Validation:** Amir Bitran, William M. Jacobs, Eugene Shakhnovich.

**Visualization:** Amir Bitran.

**Writing – original draft:** Amir Bitran.

**Writing – review & editing:** Amir Bitran, William M. Jacobs, Eugene Shakhnovich.

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
