## [Decision Letter · Decision Letter 0]

26 Sep 2020

Dear Mr Bitran,

Thank you very much for submitting your manuscript "DBFOLD: An efficient algorithm for computing folding pathways of complex proteins" for consideration at PLOS Computational Biology. As with all papers reviewed by the journal, your manuscript was reviewed by members of the editorial board and by several independent reviewers. The reviewers appreciated the attention to an important topic. Based on the reviews, we are likely to accept this manuscript for publication, providing that you modify the manuscript according to the review recommendations.

Comments from a prior reviewer are below.  Based on additional review of the manuscript by the editorial team and discussion with reviewers, we have the additional comment and requested revision:

We thank the authors for their changes to the manuscript.  We note that the systems added to the resubmitted manuscript are not novel but have been published by the authors in their prior PNAS paper.  (To be completely forthright, given the advertised computational efficiency of this approach, we would have hoped for additional protein systems.)  Since this makes the paper more a validation study than a new results study, we would suggest that the title be prefixed with "Validation of" to read "Validation of DBFOLD" before the colon.

Sincerely,

Peter M Kasson

Associate Editor

PLOS Computational Biology

Nir Ben-Tal

Deputy Editor

PLOS Computational Biology

[LINK]

Reviewer's Responses to Questions

**Comments to the Authors:**

Reviewer #1: In the revised manuscript the authors have done a good job revising the manuscript that has greatly improved.

I have just a comment about the discussion of protein G vs Protein L folding pathway, which I found a bit too qualitative. The phi-values offer a rather indirect structural information of folding pathways and transition states, and it is possible that the simulation results may actually be consistent with the experiment. To verify this, it would be necessary to compare to the raw data, ie.e folding rates and stabilities of mutants. There are ways to do this. The most accurate would be directly reproducing experimental rates and stabilities by simulation of the mutants, but this is computationally expensive. Alternatively, it may be possible to use approximate methods based on determination of the transition state for folding or of an approximate free energy landscape to infer the effect of mutations on folding rates and stabilities. I am not arguing that the authors must do this at this stage, but it may be worth at least briefly mentioning this point.

**Have all data underlying the figures and results presented in the manuscript been provided?**

Reviewer #1: Yes

PLOS authors have the option to publish the peer review history of their article (what does this mean?). If published, this will include your full peer review and any attached files.

Reviewer #1: **Yes: **Stefano Piana-Agostinetti
---

## [Editor Report · Decision Letter 1]

17 Oct 2020

Dear Mr Bitran,

We are pleased to inform you that your manuscript 'Validation of DBFOLD: An efficient algorithm for computing folding pathways of complex proteins' has been provisionally accepted for publication in PLOS Computational Biology.

Best regards,

Peter M Kasson

Associate Editor

PLOS Computational Biology

Nir Ben-Tal

Deputy Editor

PLOS Computational Biology

---

## [Editor Report · Acceptance letter]

10 Nov 2020

PCOMPBIOL-D-20-01549R1 

Validation of DBFOLD: An efficient algorithm for computing folding pathways of complex proteins

Dear Dr Bitran,

I am pleased to inform you that your manuscript has been formally accepted for publication in PLOS Computational Biology. Your manuscript is now with our production department and you will be notified of the publication date in due course.

With kind regards,

Nicola Davies
